# Deep Bayesian Bandits Showdown
## An Empirical Comparison of Bayesian Deep Networks for Thompson Sampling

**Carlos Riquelme***
Google Brain
rikel@google.com

**George Tucker**
Google Brain
gjt@google.com

**Jasper Snoek**
Google Brain
jsnoek@google.com

## Abstract

Recent advances in deep reinforcement learning have made significant strides in performance on applications such as Go and Atari games. However, developing practical methods to balance exploration and exploitation in complex domains remains largely unsolved. Thompson Sampling and its extension to reinforcement learning provide an elegant approach to exploration that only requires access to posterior samples of the model. At the same time, advances in approximate Bayesian methods have made posterior approximation for flexible neural network models practical. Thus, it is attractive to consider approximate Bayesian neural networks in a Thompson Sampling framework. To understand the impact of using an approximate posterior on Thompson Sampling, we benchmark well-established and recently developed methods for approximate posterior sampling combined with Thompson Sampling over a series of contextual bandit problems. We found that many approaches that have been successful in the supervised learning setting underperformed in the sequential decision-making scenario. In particular, we highlight the challenge of adapting slowly converging uncertainty estimates to the online setting.

## 1 Introduction

Recent advances in reinforcement learning have sparked renewed interest in sequential decision making with deep neural networks. Neural networks have proven to be powerful and flexible function approximators, allowing one to learn mappings directly from complex states (e.g., pixels) to estimates of expected return. While such models can be accurate on data they have been trained on, quantifying model uncertainty on new data remains challenging. However, having an understanding of what is not yet known or well understood is critical to some central tasks of machine intelligence, such as effective exploration for decision making.

A fundamental aspect of sequential decision making is the exploration-exploitation dilemma: in order to maximize cumulative reward, agents need to trade-off what is expected to be best at the moment, (i.e., exploitation), with potentially sub-optimal exploratory actions. Solving this trade-off in an efficient manner to maximize cumulative reward is a significant challenge as it requires uncertainty estimates. Furthermore, exploratory actions should be coordinated throughout the entire decision making process, known as deep exploration, rather than performed independently at each state.

Thompson Sampling (Thompson, 1933) and its extension to reinforcement learning, known as Posterior Sampling, provide an elegant approach that tackles the exploration-exploitation dilemma by maintaining a posterior over models and choosing actions in proportion to the probability that they are optimal. Unfortunately, maintaining such a posterior is intractable for all but the simplest models. As such, significant effort has been dedicated to approximate Bayesian methods for deep neural networks. These range from variational methods (Graves, 2011; Blundell et al., 2015; Kingma et al., 2015) to stochastic minibatch Markov Chain Monte Carlo (Neal, 1994; Welling & Teh, 2011; Li et al., 2016; Ahn et al., 2012; Mandt et al., 2016), among others. Because the exact posterior is intractable, evaluating these approaches is hard. Furthermore, these methods are rarely compared on benchmarks that measure the quality of their estimates of uncertainty for downstream tasks.

---

*Google AI Resident

To address this challenge, we develop a benchmark for exploration methods using deep neural networks. We compare a variety of well-established and recent Bayesian approximations under the lens of Thompson Sampling for contextual bandits, a classical task in sequential decision making. All code and implementations to reproduce the experiments will be available open-source, to provide a reproducible benchmark for future development. [1]

Exploration in the context of reinforcement learning is a highly active area of research. Simple strategies such as epsilon-greedy remain extremely competitive (Mnih et al., 2015; Schaul et al., 2016). However, a number of promising techniques have recently emerged that encourage exploration though carefully adding random noise to the parameters (Plappert et al., 2017; Fortunato et al., 2017; Gal & Ghahramani, 2016) or bootstrap sampling (Osband et al., 2016) before making decisions. These methods rely explicitly or implicitly on posterior sampling for exploration.

In this paper, we investigate how different posterior approximations affect the performance of Thompson Sampling from an empirical standpoint. For simplicity, we restrict ourselves to one of the most basic sequential decision making scenarios: that of contextual bandits.

No single algorithm bested the others in every bandit problem, however, we observed some general trends. We found that dropout, injecting random noise, and bootstrapping did provide a strong boost in performance on some tasks, but was not able to solve challenging synthetic exploration tasks. Other algorithms, like Variational Inference, Black Box $\alpha$-divergence, and minibatch Markov Chain Monte Carlo approaches, strongly couple their complex representation and uncertainty estimates. This proves problematic when decisions are made based on partial optimization of both, as online scenarios usually require. On the other hand, making decisions according to a Bayesian linear regression on the representation provided by the last layer of a deep network offers a robust and easy-to-tune approach. It would be interesting to try this approach on more complex reinforcement learning domains.

In Section 2 we discuss Thompson Sampling, and present the contextual bandit problem. The different algorithmic approaches that approximate the posterior distribution fed to Thompson Sampling are introduced in Section 3, while the linear case is described in Section 4. The main experimental results are presented in Section 5, and discussed in Section 6. Finally, Section 7 concludes.

## 2 DECISION-MAKING VIA THOMPSON SAMPLING

The contextual bandit problem works as follows. At time $t = 1, \ldots, n$ a new context $X_t \in \mathbf{R}^d$ arrives and is presented to algorithm $\mathcal{A}$. The algorithm —based on its internal model and $X_t$— selects one of the $k$ available actions, $a_t$. Some reward $r_t = r_t(X_t, a_t)$ is then generated and returned to the algorithm, that may update its internal model with the new data. At the end of the process, the reward for the algorithm is given by $r = \sum_{t=1}^{n} r_t$, and cumulative regret is defined as $R_{\mathcal{A}} = \mathbf{E}[r^* - r]$, where $r^*$ is the cumulative reward of the optimal policy (i.e., the policy that always selects the action with highest expected reward given the context). The goal is to minimize $R_{\mathcal{A}}$.

The main research question we address in this paper is how approximated model posteriors affect the performance of decision making via Thompson Sampling (Algorithm 1) in contextual bandits. We study a variety of algorithmic approaches to approximate a posterior distribution, together with different empirical and synthetic data problems that highlight several aspects of decision making. We consider distributions $\pi$ over the space of parameters that completely define a problem instance $\theta \in \Theta$. For example, $\theta$ could encode the reward distributions of a set of arms in the multi-armed bandit scenario, or –more generally– all the parameters of an MDP in reinforcement learning.

Thompson Sampling is a classic algorithm (Thompson, 1933) which requires only that one can sample from the posterior distribution over plausible problem instances (for example, values or rewards). At each round, it draws a sample and takes a greedy action under the optimal policy for the sample. The posterior distribution is then updated after the result of the action is observed. Thompson Sampling has been shown to be extremely effective for bandit problems both in practice (Chapelle & Li, 2011; Granmo, 2010) and theory (Agrawal & Goyal, 2012). It is especially appealing for deep neural networks as one rarely has access to the full posterior but can often approximately sample from it.

---

[1] Available in Python and Tensorflow at https://sites.google.com/site/deepbayesianbandits/.

---

**Algorithm 1** Thompson Sampling

---

1: *Input:* Prior distribution over models, $\pi_0 : \theta \in \Theta \rightarrow [0, 1]$.
2: **for** time $t = 0, \dots, N$ **do**
3:      Observe context $X_t \in \mathrm{R}^d$.
4:      Sample model $\theta_t \sim \pi_t$.
5:      Compute $a_t = \mathrm{BestAction}(X_t, \theta_t)$.
6:      Select action $a_t$ and observe reward $r_t$.
7:      Update posterior distribution $\pi_{t+1}$ with $(X_t, a_t, r_t)$.

---

In the following sections we rely on the idea that, if we had access to the actual posterior $\pi_t$ given the observed data at all times $t$, then choosing actions using Thompson Sampling would lead to near-optimal cumulative regret or, more informally, to good performance. It is important to remark that in some problems this is not necessarily the case; for example, when actions that have no chance of being optimal still convey useful information about other actions. Thompson Sampling (or UCB approaches) would never select such actions, even if they are worth their cost (Russo & Van Roy, 2014). In addition, Thompson Sampling does *not* take into account the time horizon where the process ends, and if known, exploration efforts should be tuned accordingly (Russo et al., 2017). Nonetheless, under the assumption that very accurate posterior approximations lead to efficient decisions, the question is: what happens when the approximations are not so accurate? In some cases, the mismatch in posteriors may not hurt in terms of decision making, and we will still end up with good decisions. Unfortunately, in other cases, this mismatch together with its induced feedback loop will degenerate in a significant loss of performance. We would like to understand the main aspects that determine which way it goes. This is an important practical question as, in large and complex systems, computational sacrifices and statistical assumptions are made to favor simplicity and tractability. But, what is their impact?

## 3 ALGORITHMS

In this section, we describe the different algorithmic design principles that we considered in our simulations of Section 5. These algorithms include linear methods, Neural Linear and Neural Greedy, variational inference, expectation-propagation, dropout, Monte Carlo methods, bootstrapping, direct noise injection, and Gaussian Processes. In Figure 6 in the appendix, we visualize the posteriors of the nonlinear algorithms on a synthetic one dimensional problem.

**Linear Methods** We apply well-known closed-form updates for Bayesian linear regression for exact posterior inference in linear models (Bishop, 2006). We provide the specific formulas below, and note that they admit a computationally-efficient online version. We consider exact linear posteriors as a baseline; i.e., these formulas compute the posterior when the data was generated according to $Y = X^T \beta + \epsilon$ where $\epsilon \sim \mathcal{N}(0, \sigma^2)$, and $Y$ represents the reward. Importantly, we model the *joint* distribution of $\beta$ and $\sigma^2$ for each action. Sequentially estimating the noise level $\sigma^2$ for each action allows the algorithm to adaptively improve its understanding of the volume of the hyperellipsoid of plausible $\beta$'s; in general, this leads to a more aggressive initial exploration phase (in both $\beta$ and $\sigma^2$).

The posterior at time $t$ for action $i$, after observing $X, Y$, is $\pi_t(\beta, \sigma^2) = \pi_t(\beta \mid \sigma^2) \, \pi_t(\sigma^2)$, where we assume $\sigma^2 \sim \mathrm{IG}(a_t, b_t)$, and $\beta \mid \sigma^2 \sim \mathcal{N}(\mu_t, \sigma^2 \, \Sigma_t)$, an Inverse Gamma and Gaussian distribution, respectively. Their parameters are given by

$$\Sigma_t = \left( X^T X + \Lambda_0 \right)^{-1}, \qquad \mu_t = \Sigma_t \left( \Lambda_0 \mu_0 + X^T Y \right), \qquad (1)$$

$$a_t = a_0 + t/2, \qquad b_t = b_0 + \frac{1}{2} \left( Y^T Y + \mu_0^T \Sigma_0 \mu_0 - \mu_t^T \Sigma_t^{-1} \mu_t \right). \qquad (2)$$

We set the prior hyperparameters to $\mu_0 = 0$, and $\Lambda_0 = \lambda \, \mathrm{Id}$, while $a_0 = b_0 = \eta > 1$. It follows that initially, for $\sigma_0^2 \sim IG(\eta, \eta)$, we have the prior $\beta \mid \sigma_0^2 \sim \mathcal{N}(0, \sigma_0^2/\lambda \, \mathrm{Id})$, where $\mathbf{E}[\sigma_0^2] = \eta/(\eta - 1)$. Note that we independently model and regress each action's parameters, $\beta_i, \sigma_i^2$ for $i = 1, \dots, k$.

We consider two approximations to (1) motivated by function approximators where $d$ is large. While posterior distributions or confidence ellipsoids should capture dependencies across parameters as shown above (say, a dense $\Sigma_t$), in practice, computing the correlations across all pairs of parameters is

too expensive, and diagonal covariance approximations are common. For linear models it may still be feasible to exactly compute (1), whereas in the case of Bayesian neural networks, unfortunately, this may no longer be possible. Accordingly, we study two linear approximations where $\Sigma_t$ is diagonal. Our goal is to understand the impact of such approximations in the simplest case, to properly set our expectations for the loss in performance of equivalent approximations in more complex approaches, like mean-field variational inference or Stochastic Gradient Langevin Dynamics.

Assume for simplicity the noise standard deviation is known. In Figure 2a, for $d = 2$, we see the posterior distribution $\beta_t \sim \mathcal{N}(\mu_t, \Sigma_t)$ of a linear model based on (1), in green, together with two diagonal approximations. Each approximation tries to minimize a different objective. In blue, the *PrecisionDiag* posterior approximation finds the diagonal $\hat{\Sigma} \in \mathbf{R}^{d \times d}$ minimizing $\mathrm{KL}(\mathcal{N}(\mu_t, \hat{\Sigma}) \ || \ \mathcal{N}(\mu_t, \Sigma_t))$, like in mean-field variational inference. In particular, $\hat{\Sigma} = \mathrm{Diag}(\Sigma_t^{-1})^{-1}$. On the other hand, in orange, the *Diag* posterior approximation finds the diagonal matrix $\bar{\Sigma}$ minimizing $\mathrm{KL}(\mathcal{N}(\mu_t, \Sigma_t) \ || \ \mathcal{N}(\mu_t, \bar{\Sigma}))$ instead. In this case, the solution is simply $\bar{\Sigma} = \mathrm{Diag}(\Sigma_t)$.

We add linear baselines that do *not* model the uncertainty in the action noise $\sigma^2$. In addition, we also consider simple greedy and epsilon greedy linear baselines (i.e., not based on Thompson Sampling).

**Neural Linear** The main problem linear algorithms face is their lack of representational power, which they complement with accurate uncertainty estimates. A natural attempt at getting the best of both worlds consists in performing a Bayesian linear regression on top of the representation of the last layer of a neural network, similarly to Snoek et al. (2015). The predicted value $v_i$ for each action $a_i$ is given by $v_i = \beta_i^T z_x$, where $z_x$ is the output of the last hidden layer of the network for context $x$. While linear methods directly try to regress values $v$ on $x$, we can independently train a deep net to learn a representation $z$, and then use a Bayesian linear regression to regress $v$ on $z$, obtain uncertainty estimates on the $\beta$'s, and make decisions accordingly via Thompson Sampling. Note that we do not explicitly consider the weights of the linear output layer of the network to make decisions; further, the network is *only* used to find good representations $z$. In addition, we can update the network and the linear regression at different time-scales. It makes sense to keep an exact linear regression (as in (1) and (2)) at all times, adding each new data point as soon as it arrives. However, we only update the network after a number of points have been collected. In our experiments, after updating the network, we perform a forward pass on all the training data to obtain $z_x$, which is then fed to the Bayesian regression. In practice this may be too expensive, and $z$ could be updated periodically with online updates on the regression. We call this algorithm *Neural Linear*.

**Neural Greedy** We refer to the algorithm that simply trains a neural network and acts greedily (i.e., takes the action whose predicted score for the current context is highest) as **RMS**, as we train it using the RMSProp optimizer. This is our non-linear baseline, and we tested several versions of it (based on whether the training step was decayed, reset to its initial value for each re-training or not, and how long the network was trained for). We also tried the $\epsilon$-greedy version of the algorithm, where a random action was selected with probability $\epsilon$ for some decaying schedule of $\epsilon$.

**Variational Inference** Variational approaches approximate the posterior by finding a distribution within a tractable family that minimizes the KL divergence to the posterior (Hinton & Van Camp, 1993). These approaches formulate and solve an optimization problem, as opposed, for example, to sampling methods like MCMC (Jordan et al., 1999; Wainwright et al., 2008). Typically (and in our experiments), the posterior is approximated by a mean-field or factorized distribution where strong independence assumptions are made. For instance, each neural network weight can be modeled via a –conditionally independent– Gaussian distribution whose mean and variance are estimated from data. Recent advances have scaled these approaches to estimate the posterior of neural networks with millions of parameters (Blundell et al., 2015). A common criticism of variational inference is that it underestimates uncertainty (e.g., (Bishop, 2006)), which could lead to under-exploration.

**Expectation-Propagation** The family of expectation-propagation algorithms (Opper & Winther, 2000; Minka, 2001b;a) is based on the message passing framework (Pearl, 1986). They iteratively approximate the posterior by updating a single approximation factor (or *site*) at a time, which usually corresponds to the likelihood of one data point. The algorithm sequentially minimizes a set of local KL divergences, one for each site. Most often, and for computational reasons, likelihoods are chosen to lie in the exponential family. In this case, the minimization corresponds to moment matching. See Gelman et al. (2014) for further details. We focus on methods that directly optimize the *global* EP objective via stochastic gradient descent, as, for instance, Power EP (Minka, 2004). In particular, in

this work, we implement the black-box $\alpha$-divergence minimization algorithm (Hernández-Lobato et al., 2016), where local parameter sharing is applied to the Power EP energy function. Note that different values of $\alpha \in \mathbf{R}\backslash\{0\}$ correspond to common algorithms: $\alpha = 1$ to EP, and $\alpha \rightarrow 0$ to Variational Bayes. The optimal $\alpha$ value is problem-dependent (Hernández-Lobato et al., 2016).

**Dropout** Dropout is a training technique where the output of each neuron is independently zeroed out with probability $p$ at each forward pass (Srivastava et al., 2014). Once the network has been trained, dropout can still be used to obtain a distribution of predictions for a specific input. Following the best action with respect to the *random* dropout prediction can be interpreted as an implicit form of Thompson sampling. Dropout can be seen as optimizing a variational objective (Kingma et al., 2015; Gal & Ghahramani, 2016; Hron et al., 2017).

**Monte Carlo** Monte Carlo sampling remains one of the simplest and reliable tools in the Bayesian toolbox. Rather than parameterizing the full posterior, Monte Carlo methods estimate the posterior through drawing samples. This is naturally appealing for highly parameterized deep neural networks for which the posterior is intractable in general and even simple approximations such as multivariate Gaussian are too expensive (i.e. require computing and inverting a covariance matrix over all parameters). Among Monte Carlo methods, Hamiltonian Monte Carlo (Neal, 1994) (HMC) is often regarded as a gold standard algorithm for neural networks as it takes advantage of gradient information and momentum to more effectively draw samples. However, it remains unfeasible for larger datasets as it involves a Metropolis accept-reject step that requires computing the log likelihood over the whole data set. A variety of methods have been developed to approximate HMC using mini-batch stochastic gradients. These Stochastic Gradient Langevin Dynamics (SGLD) methods (Neal, 1994; Welling & Teh, 2011) add Gaussian noise to the model gradients during stochastic gradient updates in such a manner that each update results in an approximate sample from the posterior. Different strategies have been developed for augmenting the gradients and noise according to a preconditioning matrix. Li et al. (2016) show that a preconditioner based on the RMSprop algorithm performs well on deep neural networks. Patterson & Teh (2013) suggested using the Fisher information matrix as a preconditioner in SGLD. Unfortunately the approximations of SGLD hold only if the learning rate is asymptotically annealed to zero. Ahn et al. (2012) introduced Stochastic Gradient Fisher Scoring to elegantly remove this requirement by preconditioning according to the Fisher information (or a diagonal approximation thereof). Mandt et al. (2016) develop methods for approximately sampling from the posterior using a constant learning rate in stochastic gradient descent and develop a prescription for a stable version of SGFS. We evaluate the diagonal-SGFS and constant-SGD algorithms from Mandt et al. (2016) in this work. Specifically for constant-SGD we use a constant learning rate for stochastic gradient descent, where the learning rate $\epsilon$ is given by $\epsilon = 2\frac{S}{N}\mathbf{BB}^T$ where $S$ is the batch size, $N$ the number of data points and $\mathbf{BB}^T$ is an online average of the diagonal empirical Fisher information matrix. For Stochastic Gradient Fisher Scoring we use the following stochastic gradient update for the model parameters $\theta$ at step $t$:

$$\theta_{t+1} = \theta_t - \epsilon \, \mathbf{H} \, g(\theta_t) + \sqrt{\epsilon} \, \mathbf{H} \, \mathbf{E} \, \nu, \qquad \nu \sim \mathcal{N}(0, I) \tag{3}$$

where we take the noise covariance $\mathbf{EE}^T$ to also be $\mathbf{BB}^T$ and $\mathbf{H} = \frac{2}{N}(\epsilon\mathbf{BB}^T + \mathbf{EE}^\mathbf{T})^{-1}$.

**Bootstrap** A simple empirical approach to approximate the sampling distribution of any estimator is the Bootstrap (Efron, 1982). The main idea is to simultaneously train $q$ models, where each model $i$ is based on a different dataset $D_i$. When all the data $D$ is available in advance, $D_i$ is typically created by sampling $|D|$ elements from $D$ at random with replacement. In our case, however, the data grows one example at a time. Accordingly, we set a parameter $p \in (0, 1]$, and append the new datapoint to each $D_i$ independently at random with probability $p$. In order to emulate Thompson Sampling, we sample a model uniformly at random (i.e., with probability $p_i = 1/q$.) and take the action predicted to be best by the sampled model. We mainly tested cases $q = 5, 10$ and $p = 0.8, 1.0$, with neural network models. Note that even when $p = 1$ and the datasets are identical, the random initialization of each network, together with the randomness from SGD, lead to different predictions.

**Direct Noise Injection** Parameter-Noise (Plappert et al., 2017) is a recently proposed approach for exploration in deep RL that has shown promising results. The training updates for the network are unchanged, but when selecting actions, the network weights $\theta$ are perturbed with isotropic Gaussian noise. Crucially, the network uses layer normalization (Ba et al., 2016), which ensures that all weights are on the same scale. The magnitude of the Gaussian noise is adjusted so that the overall effect of the perturbations is similar in scale to $\epsilon$-greedy with a linearly decaying schedule (see (Plappert et al.,

2017) for details). Because the perturbations are done on the model parameters, we might hope that the actions produced by the perturbations are more sensible than $\epsilon$-greedy.

**Bayesian Non-parametric** Gaussian processes (Rasmussen & Williams, 2005) are a gold-standard method for modeling distributions over non-linear continuous functions. It can be shown that, in the limit of infinite hidden units and under a Gaussian prior, a Bayesian neural network converges to a Gaussian process (Neal, 1994). As such, GPs would appear to be a natural baseline. Unfortunately, standard GPs computationally scale cubically in the number of observations, limiting their applicability to relatively small datasets. There are a wide variety of methods to approximate Gaussian processes using, for example, pseudo-observations (Snelson & Ghahramani, 2006) or variational inference (Titsias, 2009). We implemented both standard and sparse GPs but only report the former due to similar performance. For the standard GP, due to the scaling issue, we stop adding inputs to the GP after 1000 observations. This performed significantly better than randomly sampling inputs. Our implementation is a multi-task Gaussian process (Bonilla et al., 2008) with a linear and Matern $3/2$ product kernel over the inputs and an exponentiated quadratic kernel over latent vectors for the different tasks. The hyperparameters of this model and the latent task vectors are optimized over the GP marginal likelihood. This allows the model to learn correlations between the outputs of the model. Specifically, the covariance function $K(\cdot)$ of the GP is given by:

$$K(\mathbf{x}^k, \hat{\mathbf{x}}^l) = k_{\text{matern}}(\mathbf{x}^k, \hat{\mathbf{x}}^l) \cdot k_{\text{lin}}(\mathbf{x}^k, \hat{\mathbf{x}}^l) \cdot k_{\text{task}}(\mathbf{v}^k, \mathbf{v}^l) \tag{4}$$

$$k_{\text{matern}}(\mathbf{x}^k, \hat{\mathbf{x}}^l) = \alpha(1 + \sqrt{3}r_{\lambda_m}(\mathbf{x}, \hat{\mathbf{x}})) \exp(-\sqrt{3}r_{\lambda_m}(\mathbf{x}, \hat{\mathbf{x}}))) \tag{5}$$

$$k_{\text{lin}}(\mathbf{x}^k, \hat{\mathbf{x}}^l) = \beta(\mathbf{x} \oslash \lambda_l)(\hat{\mathbf{x}} \oslash \mathbf{\lambda_l})^T \tag{6}$$

and the task kernel between tasks $t$ and $l$ are $k_t(\mathbf{x}^k, \hat{\mathbf{x}}^l) = \exp(-r_{\lambda_m}(\mathbf{v}^k, \hat{\mathbf{v}}^l))^2)$ where $\mathbf{v}^l$ indexes the latent vector for task $l$ and $r_\lambda(\mathbf{x}, \hat{\mathbf{x}}) = |(\mathbf{x} \oslash \lambda) - (\hat{\mathbf{x}} \oslash \lambda)|$. The length-scales, $\lambda_m$ and $\lambda_l$, and amplitude parameters $\alpha, \beta$ are optimized via the log marginal likelihood. For the sparse version we used a Sparse Variational GP (Hensman et al., 2015) with the same kernel and with 300 inducing points, trained via minibatch stochastic gradient descent (Matthews et al., 2017).

## 4 FEEDBACK LOOP IN THE LINEAR CASE

In this section, we illustrate some of the subtleties that arise when uncertainty estimates drive sequential decision-making using simple linear examples.

There is a fundamental difference between *static* and *dynamic* scenarios. In a static scenario, e.g. supervised learning, we are given a model family $\Theta$ (like the set of linear models, trees, or neural networks with specific dimensions), a prior distribution $\pi_0$ over $\Theta$, and some observed data $\mathbf{D}$ that —importantly— is assumed i.i.d. Our goal is to return an approximate posterior distribution: $\tilde{\pi} \approx \pi = \mathbf{P}(\theta \mid \mathbf{D})$. We define the quality of our approximation by means of some distance $d(\tilde{\pi}, \pi)$.

On the other hand, in *dynamic* settings, our estimate at time $t$, say $\tilde{\pi}_t$, will be used via some mechanism $\mathbf{M}$, in this case Thompson sampling, to collect the next data-point, which is then appended to $\mathbf{D}_t$. In this case, the data-points in $\mathbf{D}_t$ are no longer independent. $\mathbf{D}_t$ will now determine two distributions: the posterior given the data that was actually observed, $\pi_{t+1} = \mathbf{P}(\theta \mid \mathbf{D}_t)$, and our new estimate $\tilde{\pi}_{t+1}$. When the goal is to make good sequential decisions in terms of cumulative regret, the distance $d(\tilde{\pi}_t, \pi_t)$ is in general no longer a definitive proxy for performance. For instance, a poorly-approximated decision boundary could lead an algorithm, based on $\tilde{\pi}$, to get stuck repeatedly selecting a single sub-optimal action $a$. After collecting lots of data for that action, $\tilde{\pi}_t$ and $\pi_t$ could start to agree (to their capacity) on the models that explain what was observed for $a$, while both would stick to something close to the prior regarding the other actions. At that point, $d(\tilde{\pi}_t, \pi_t)$ may show relatively little disagreement, but the regret would already be terrible.

Let $\pi_t^*$ be the posterior distribution $\mathbf{P}(\theta \mid \mathbf{D}_t)$ under Thompson Sampling's assumption, that is, data was always collected according to $\pi_j^*$ for $j < t$. We follow the idea that $\tilde{\pi}_t$ being close to $\pi_t^*$ for all $t$ leads to strong performance. However, this concept is difficult to formalize: once different decisions are made, data for different actions is collected and it is hard to compare posterior distributions.

We illustrate the previous points with a simple example, see Figure 1. Data is generated according to a bandit with $k = 6$ arms. For a given context $X \sim \mathcal{N}(\mu, \Sigma)$, the reward obtained by pulling arm $i$ follows a linear model $r_{i,X} = X^T \beta_i + \epsilon$ with $\epsilon \sim \mathcal{N}(0, \sigma_i^2)$. The posterior distribution over

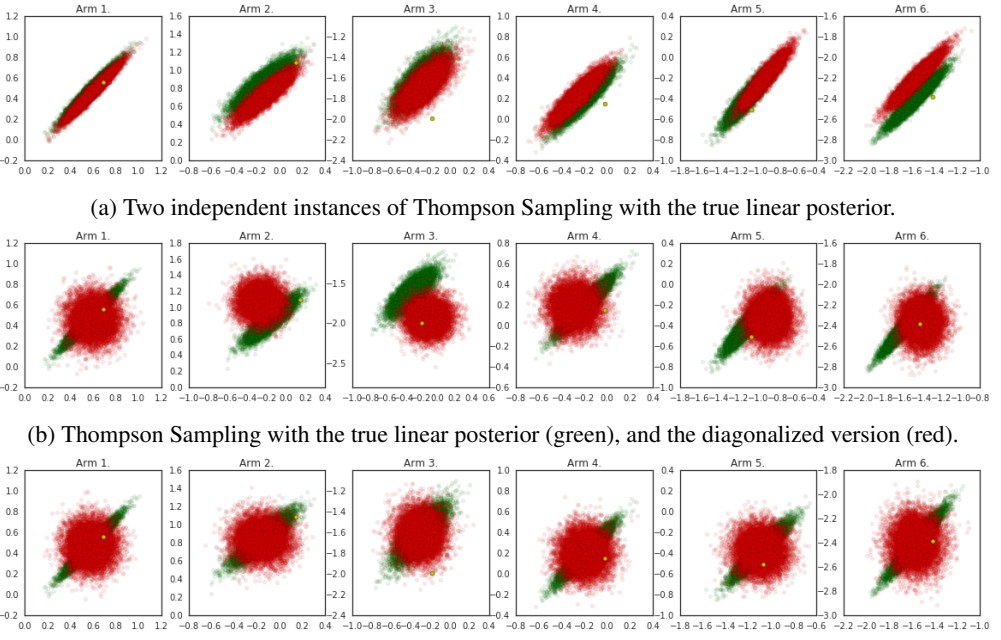

(a) Two independent instances of Thompson Sampling with the true linear posterior.

(b) Thompson Sampling with the true linear posterior (green), and the diagonalized version (red).

(c) Linear posterior versus diagonal posterior fitted to the data collected by the former.

Figure 1: Visualizations of the posterior approximations in a linear example.

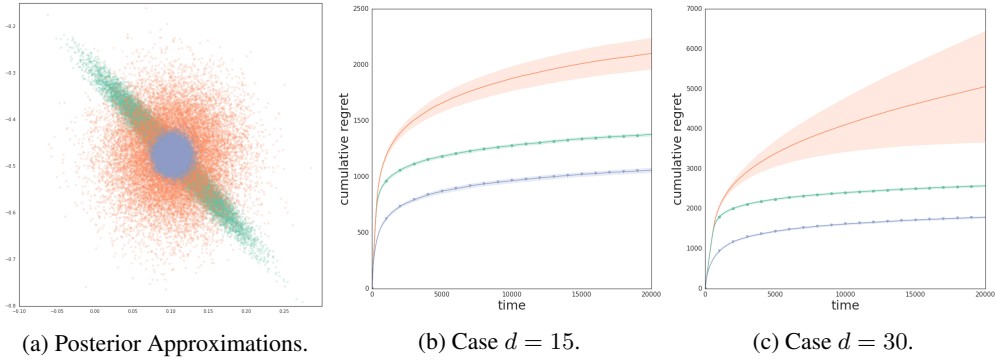

(a) Posterior Approximations.       (b) Case $d = 15$.       (c) Case $d = 30$.

Figure 2: The impact on regret of different approximated posteriors. We show (green) the actual linear posterior, (orange) the diagonal posterior approximation and (blue) the precision approximation in 2a. In 2b and 2c we visualize the impact of the approximations on cumulative regret.

$\beta_i \in \mathrm{R}^d$ can be exactly computed using the standard Bayesian linear regression formulas presented in Section 3. We set the contextual dimension $d = 20$, and the prior to be $\beta \sim \mathcal{N}(0, \lambda\, \mathrm{I}_d)$, for $\lambda > 0$.

In Figure 1, we show the posterior distribution for two dimensions of $\beta_i$ for each arm $i$ after $n = 500$ pulls. In particular, in Figure 1a, two independent runs of Thompson Sampling with their posterior distribution are displayed in red and green. While strongly aligned, the estimates for some arms disagree (especially for arms that are best only for a *small* fraction of the contexts, like Arm 2 and 3, where fewer data-points are available). In Figure 1b, we also consider Thompson Sampling with an approximate posterior with diagonal covariance matrix, *Diag* in red, as defined in Section 3. Each algorithm collects its own data based on its current posterior (or approximation). In this case, the posterior disagreement after $n = 500$ decisions is certainly stronger. However, as shown in Figure 1c, if we computed the approximate posterior with a diagonal covariance matrix based on the data collected *by the actual posterior*, the disagreement would be reduced as much as possible within the approximation capacity (i.e., it still cannot capture correlations in this case). Figure 1b shows then the effect of the feedback loop. We look next at the impact that this mismatch has on regret.

We illustrate with a similar example how inaccurate posteriors sometimes lead to quite different behaviors in terms of regret. In Figure 2a, we see the posterior distribution $\beta \sim \mathcal{N}(\mu, \Sigma)$ of a linear model in green, together with the two diagonal linear approximations introduced in Section 3: the Diag (in orange) and the PrecisionDiag (in blue) approximations, respectively. We now assume there are $k$ linear arms, $\beta_i \in \mathbf{R}^d$ for $i = 1, \ldots, k$, and decisions are made according to the posteriors in Figure 2a. In Figures 2b and 2c we plot the regret of Thompson Sampling when there are $k = 20$ arms, for both $d = 15$ and $d = 30$. We see that, while the PrecisionDiag approximation does even outperform the actual posterior, the diagonal covariance approximation truly suffers poor regret when we increase the dimension $d$, as it is heavily penalized by *simultaneously* over-exploring in a large number of dimensions and repeatedly acting according to implausible models.

## 5 EMPIRICAL EVALUATION

In this section, we present the simulations and outcomes of several synthetic and real-world data bandit problems with each of the algorithms introduced in Section 3. In particular, we first explain how the simulations were set up and run, and the metrics we report. We then split the experiments according to how data was generated, and the underlying models fit by the algorithms from Section 3.

### 5.1 THE EXPERIMENTAL FRAMEWORK

We run the contextual bandit experiments as described at the beginning of Section 2, and discuss below some implementation details of both experiments and algorithms. A detailed summary of the key parameters used for each algorithm can be found in Table 2 in the appendix.

**Neural Network Architectures** All algorithms based on neural networks as function approximators share the same architecture. In particular, we fit a simple fully-connected feedforward network with two hidden layers with 100 units each and ReLu activations. The input of the network has dimension $d$ (same as the contexts), and there are $k$ outputs, one per action. Note that for each training point $(X_t, a_t, r_t)$ only one action was observed (and algorithms usually only take into account the loss corresponding to the prediction for the observed action).

**Updating Models** A key question is how often and for how long models are updated. Ideally, we would like to train after each new observation and for as long as possible. However, this may limit the applicability of our algorithms in online scenarios where decisions must be made immediately. We update linear algorithms after each time-step by means of (1) and (2). For neural networks, the default behavior was to train for $t_s = 20$ or 100 mini-batches every $t_f = 20$ timesteps. [2] The size of each mini-batch was 512. We experimented with increasing values of $t_s$, and it proved essential for some algorithms like variational inference approaches. See the details in Table 2.

**Metrics** We report two metrics: cumulative regret and simple regret. We approximate the latter as the mean cumulative regret in the last 500 time-steps, a proxy for the quality of the final policy (see further discussion on pure exploration settings, Bubeck et al. (2009)). Cumulative regret is computed based on the best expected reward, as is standard. For most real datasets (Statlog, Covertype, Jester, Adult, Census, and Song), the rewards were deterministic, in which case, the definition of regret also corresponds to the highest *realized* reward (i.e., possibly leading to a hard task, which helps to understand why in some cases all regrets look linear). We reshuffle the order of the contexts, and rerun the experiment 50 times to obtain the cumulative regret distribution and report its statistics.

**Hyper-Parameter Tuning** Deep learning methods are known to be very sensitive to the selection of a wide variety of hyperparameters, and many of the algorithms presented are no exception. Moreover, that choice is known to be highly dataset dependent. Unfortunately, in the bandits scenario, we commonly do not have access to each problem a-priori to perform tuning. For the vast majority of algorithms, we report the outcome for *three* versions of the algorithm defined as follows. First, we use one version where hyper-parameters take values we guessed to be reasonable a-priori. Then, we add two additional instances whose hyper-parameters were optimized on two different datasets via Bayesian Optimization. For example, in the case of Dropout, the former version is named Dropout, while the optimized versions are named Dropout-MR (using the Mushroom dataset) and Dropout-SL (using the Statlog dataset) respectively. Some algorithms truly benefit from hyper-parameter

---

[2]For reference, the standard strategy for Deep Q-Networks on Atari is to make one model update after every 4 actions performed (Mnih et al., 2015; Osband et al., 2016; Plappert et al., 2017; Fortunato et al., 2017).

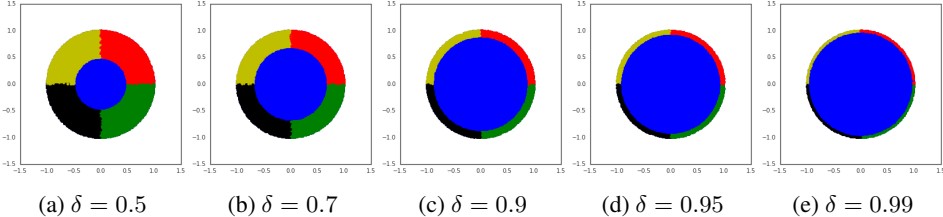

| (a) $\delta = 0.5$ | (b) $\delta = 0.7$ | (c) $\delta = 0.9$ | (d) $\delta = 0.95$ | (e) $\delta = 0.99$ |

Figure 3: Wheel bandits for increasing values of $\delta \in (0, 1)$. Optimal action for blue, red, green, black, and yellow regions, are actions 1, 2, 3, 4, and 5, respectively.

optimization, while others do not show remarkable differences in performance; the latter are more appropriate in settings where access to the real environment for tuning is not possible in advance.

**Buffer** After some experimentation, we decided not to use a data buffer as evidence of catastrophic forgetting was observed, and datasets are relatively small. Accordingly, all observations are sampled with equal probability to be part of a mini-batch. In addition, as is standard in bandit algorithms, each action was initially selected $s = 3$ times using round-robin independently of the context.

## 5.2 REAL-WORLD DATA PROBLEMS WITH NON-LINEAR MODELS

We evaluated the algorithms on a range of bandit problems created from real-world data. In particular, we test on the Mushroom, Statlog, Covertype, Financial, Jester, Adult, Census, and Song datasets (see Appendix Section A for details on each dataset and bandit problem). They exhibit a broad range of properties: small and large sizes, one dominating action versus more homogeneous optimality, learnable or little signal, stochastic or deterministic rewards, etc. For space reasons, the outcome of some simulations are presented in the Appendix. The Statlog, Covertype, Adult, and Census datasets were originally tested in Elmachtoub et al. (2017). We summarize the final cumulative regret for Mushroom, Statlog, Covertype, Financial, and Jester datasets in Table 1. In Figure 5 at the appendix, we show a box plot of the ranks achieved by each algorithm across the suite of bandit problems (see Appendix Table 6 and 7 for the full results).

## 5.3 REAL-WORLD DATA PROBLEMS WITH LINEAR MODELS

As most of the algorithms from Section 3 can be implemented for any model architecture, in this subsection we use linear models as a baseline comparison across algorithms (i.e., neural networks that contain a single linear layer). This allows us to directly compare the approximate methods against methods that can compute the exact posterior. The specific hyper-parameter configurations used in the experiments are described in Table 3 in the appendix. Datasets are the same as in the previous subsection. The cumulative and simple regret results are provided in appendix Tables 4 and 5.

## 5.4 THE WHEEL BANDIT

Some of the real-data problems presented above do not require significant exploration. We design an artificial problem where the need for exploration is smoothly parameterized. The *wheel* bandit is defined as follows (see Figure 3). Set $d = 2$, and $\delta \in (0, 1)$, the exploration parameter. Contexts are sampled uniformly at random in the unit circle in $\mathbf{R}^2$, $X \sim U(\mathbf{D})$. There are $k = 5$ possible actions. The first action $a_1$ always offers reward $r \sim \mathcal{N}(\mu_1, \sigma^2)$, independently of the context. On the other hand, for contexts such that $\|X\| \leq \delta$, i.e. inside the blue circle in Figure 3, the other four actions are equally distributed and sub-optimal, with $r \sim \mathcal{N}(\mu_2, \sigma^2)$ for $\mu_2 < \mu_1$. When $\|X\| > \delta$, we are outside the blue circle, and only one of the actions $a_2, \ldots, a_5$ is optimal depending on the sign of context components $X = (X_1, X_2)$. If $X_1, X_2 > 0$, action 2 is optimal. If $X_1 > 0, X_2 < 0$, action 3 is optimal, and so on. Non-optimal actions still deliver $r \sim \mathcal{N}(\mu_2, \sigma^2)$ in this region, except $a_1$ whose mean reward is always $\mu_1$, while the optimal action provides $r \sim \mathcal{N}(\mu_3, \sigma^2)$, with $\mu_3 \gg \mu_1$. We set $\mu_1 = 1.2, \mu_2 = 1.0, \mu_3 = 50.0$, and $\sigma = 0.01$. Note that the probability of a context randomly falling in the high-reward region is $1 - \delta^2$ (not blue). The difficulty of the problem increases with $\delta$, and we expect algorithms to get stuck repeatedly selecting action $a_1$ for large $\delta$. The problem can be easily generalized for $d > 2$. Results are shown in Table 9.

Table 1: Cumulative regret incurred by the algorithms in Section 3 on the bandits described in Section A. Results are relative to the cumulative regret of the Uniform algorithm. We report the mean and standard error of the mean over 50 trials.

| | Mean Rank | Mushroom | Statlog | Covertype | Financial | Jester |
|---|---|---|---|---|---|---|
| AlphaDivergence (1) | 47 | $54.29 \pm 0.04$ | $19.35 \pm 1.72$ | $39.42 \pm 0.50$ | $40.10 \pm 0.69$ | $72.99 \pm 0.54$ |
| AlphaDivergence | 46.6 | $54.17 \pm 0.03$ | $19.30 \pm 0.84$ | $44.31 \pm 0.77$ | $47.76 \pm 0.89$ | $71.86 \pm 0.72$ |
| AlphaDivergence-SL | 44.3 | $53.88 \pm 0.03$ | $21.12 \pm 1.14$ | $60.05 \pm 0.02$ | $70.19 \pm 3.50$ | $69.11 \pm 0.75$ |
| BBB | 39.8 | $3.57 \pm 0.20$ | $12.61 \pm 1.53$ | $58.19 \pm 2.16$ | $22.55 \pm 1.27$ | $71.43 \pm 0.67$ |
| BBB-MR | 37.5 | $10.93 \pm 2.69$ | $25.29 \pm 0.00$ | $60.92 \pm 0.55$ | $59.84 \pm 2.89$ | $65.01 \pm 0.74$ |
| BBB-SL | 34.4 | $4.08 \pm 0.19$ | $1.86 \pm 0.29$ | $38.50 \pm 0.97$ | $13.76 \pm 0.60$ | $74.70 \pm 0.68$ |
| BootstrappedNN | 22.4 | $5.60 \pm 0.60$ | $0.65 \pm 0.02$ | $26.23 \pm 0.10$ | $9.21 \pm 0.44$ | $73.38 \pm 0.62$ |
| BootstrappedNN-MR | 22.6 | $2.15 \pm 0.13$ | $1.19 \pm 0.15$ | $31.27 \pm 0.08$ | $7.72 \pm 0.28$ | $63.26 \pm 0.58$ |
| BootstrappedNN-SL | 22.9 | $3.93 \pm 0.17$ | $0.54 \pm 0.01$ | $25.53 \pm 0.05$ | $10.88 \pm 0.61$ | $70.64 \pm 0.59$ |
| Dropout | 30.5 | $5.57 \pm 1.02$ | $2.35 \pm 0.59$ | $30.65 \pm 0.23$ | $17.55 \pm 0.67$ | $66.24 \pm 0.74$ |
| Dropout-MR | 20.3 | $2.65 \pm 0.08$ | $1.30 \pm 0.32$ | $29.28 \pm 0.12$ | $10.16 \pm 0.44$ | $63.68 \pm 0.60$ |
| Dropout-SL | 26.4 | $4.39 \pm 1.02$ | $1.89 \pm 0.47$ | $26.39 \pm 0.17$ | $13.18 \pm 0.86$ | $66.90 \pm 0.80$ |
| GP | 31.75 | $11.49 \pm 0.66$ | $3.92 \pm 0.74$ | $46.25 \pm 0.75$ | $3.18 \pm 0.08$ | $74.95 \pm 0.93$ |
| NeuralLinear | 22 | $2.22 \pm 0.08$ | $0.91 \pm 0.01$ | $29.91 \pm 0.17$ | $11.44 \pm 0.11$ | $75.43 \pm 0.41$ |
| NeuralLinear-MR | 22.4 | $\mathbf{1.92 \pm 0.10}$ | $1.30 \pm 0.01$ | $28.87 \pm 0.14$ | $13.47 \pm 0.12$ | $72.75 \pm 0.50$ |
| NeuralLinear-SL | 17.4 | $2.42 \pm 0.09$ | $\mathbf{0.52 \pm 0.01}$ | $27.60 \pm 0.10$ | $9.98 \pm 0.56$ | $71.11 \pm 0.47$ |
| RMS | 28.7 | $6.68 \pm 1.52$ | $2.85 \pm 0.73$ | $27.74 \pm 0.18$ | $12.73 \pm 0.73$ | $69.93 \pm 0.56$ |
| RMS-MR | 29.8 | $4.32 \pm 1.06$ | $2.36 \pm 0.44$ | $32.46 \pm 0.57$ | $10.72 \pm 0.51$ | $68.43 \pm 0.72$ |
| RMS-SL | 30.7 | $3.29 \pm 0.16$ | $2.22 \pm 0.87$ | $28.25 \pm 0.14$ | $12.76 \pm 0.63$ | $71.50 \pm 0.49$ |
| SGFS | 30.8 | $5.99 \pm 1.02$ | $3.82 \pm 0.45$ | $36.57 \pm 0.80$ | $29.00 \pm 0.53$ | $68.02 \pm 0.63$ |
| SGFS-MR | 21.8 | $3.80 \pm 0.46$ | $1.44 \pm 0.01$ | $30.12 \pm 0.05$ | $12.49 \pm 0.71$ | $66.27 \pm 0.72$ |
| SGFS-SL | 30.9 | $2.79 \pm 0.10$ | $2.14 \pm 0.13$ | $35.27 \pm 0.27$ | $43.95 \pm 1.12$ | $73.90 \pm 1.51$ |
| ConstSGD | 36.9 | $26.37 \pm 3.14$ | $6.79 \pm 1.42$ | $\mathbf{22.47 \pm 0.78}$ | $88.16 \pm 2.80$ | $70.09 \pm 0.80$ |
| ConstSGD-MR | 29.9 | $3.10 \pm 0.08$ | $7.24 \pm 1.07$ | $23.39 \pm 0.66$ | $46.47 \pm 1.85$ | $71.25 \pm 0.61$ |
| ConstSGD-SL | 35.1 | $41.94 \pm 2.31$ | $2.96 \pm 0.79$ | $\mathbf{21.61 \pm 0.15}$ | $51.94 \pm 3.78$ | $70.24 \pm 0.95$ |
| EpsGreedyRMS | 23.6 | $4.97 \pm 1.04$ | $2.13 \pm 0.34$ | $27.42 \pm 0.13$ | $12.36 \pm 0.47$ | $69.65 \pm 0.70$ |
| EpsGreedyRMS-SL | 23.2 | $3.08 \pm 0.15$ | $1.09 \pm 0.20$ | $28.09 \pm 0.09$ | $7.93 \pm 0.39$ | $69.64 \pm 0.61$ |
| EpsGreedyRMS-MR | 24.4 | $2.44 \pm 0.15$ | $1.71 \pm 0.44$ | $30.03 \pm 0.20$ | $8.07 \pm 0.45$ | $66.18 \pm 0.57$ |
| LinDiagPost | 30.6 | $17.67 \pm 0.18$ | $51.29 \pm 0.03$ | $95.48 \pm 0.02$ | $9.59 \pm 0.05$ | $\mathbf{58.61 \pm 0.49}$ |
| LinDiagPost-MR | 37.3 | $8.64 \pm 2.33$ | $31.51 \pm 0.03$ | $65.03 \pm 0.03$ | $20.57 \pm 0.13$ | $60.62 \pm 0.49$ |
| LinDiagPost-SL | 31.5 | $14.86 \pm 2.12$ | $15.60 \pm 0.03$ | $42.72 \pm 0.03$ | $2.04 \pm 0.04$ | $59.96 \pm 0.67$ |
| LinDiagPrecPost | 15.8 | $9.48 \pm 1.59$ | $7.53 \pm 0.02$ | $34.40 \pm 0.02$ | $4.58 \pm 0.04$ | $\mathbf{58.58 \pm 0.60}$ |
| LinDiagPrecPost-MR | 26.1 | $16.21 \pm 3.14$ | $8.77 \pm 0.07$ | $35.69 \pm 0.02$ | $9.15 \pm 0.10$ | $\mathbf{59.08 \pm 0.45}$ |
| LinDiagPrecPost-SL | 16.9 | $13.17 \pm 1.73$ | $6.80 \pm 0.02$ | $33.85 \pm 0.07$ | $\mathbf{1.82 \pm 0.06}$ | $\mathbf{58.83 \pm 0.45}$ |
| LinGreedy | 25.3 | $14.28 \pm 1.99$ | $11.32 \pm 0.63$ | $35.29 \pm 0.11$ | $2.18 \pm 0.14$ | $\mathbf{59.69 \pm 0.60}$ |
| LinGreedy ($\epsilon = 0.01$) | 18.1 | $3.38 \pm 0.18$ | $10.42 \pm 0.39$ | $34.59 \pm 0.08$ | $2.94 \pm 0.12$ | $59.95 \pm 0.58$ |
| LinGreedy ($\epsilon = 0.05$) | 19.6 | $5.89 \pm 0.06$ | $12.75 \pm 0.16$ | $37.00 \pm 0.03$ | $6.57 \pm 0.11$ | $61.62 \pm 0.43$ |
| LinPost | 16.4 | $6.12 \pm 0.67$ | $7.64 \pm 0.02$ | $34.40 \pm 0.02$ | $7.26 \pm 0.05$ | $\mathbf{59.14 \pm 0.50}$ |
| LinPost-MR | 28.3 | $7.93 \pm 2.00$ | $10.31 \pm 0.03$ | $38.64 \pm 0.02$ | $15.61 \pm 0.10$ | $\mathbf{59.17 \pm 0.56}$ |
| LinPost-SL | 18.9 | $14.34 \pm 1.84$ | $6.82 \pm 0.02$ | $33.61 \pm 0.03$ | $2.50 \pm 0.03$ | $60.02 \pm 0.57$ |
| LinFullDiagPost | 33.1 | $86.80 \pm 0.13$ | $28.29 \pm 0.02$ | $73.82 \pm 0.03$ | $6.96 \pm 0.06$ | $63.22 \pm 0.61$ |
| LinFullDiagPost-MR | 28.2 | $2.39 \pm 0.08$ | $14.24 \pm 0.02$ | $37.59 \pm 0.02$ | $10.25 \pm 0.11$ | $62.87 \pm 0.42$ |
| LinFullDiagPost-SL | 27.9 | $2.24 \pm 0.10$ | $12.04 \pm 0.04$ | $37.08 \pm 0.06$ | $10.92 \pm 0.45$ | $62.56 \pm 0.51$ |
| LinFullDiagPrecPost | 14.3 | $3.47 \pm 0.36$ | $7.34 \pm 0.03$ | $34.04 \pm 0.02$ | $4.04 \pm 0.05$ | $60.63 \pm 0.44$ |
| LinFullDiagPrecPost-MR | 15.6 | $2.90 \pm 0.34$ | $7.88 \pm 0.03$ | $34.24 \pm 0.03$ | $7.74 \pm 0.06$ | $60.65 \pm 0.50$ |
| LinFullDiagPrecPost-SL | 17.7 | $2.65 \pm 0.14$ | $6.84 \pm 0.02$ | $33.97 \pm 0.06$ | $4.99 \pm 0.18$ | $60.99 \pm 0.55$ |
| LinFullPost | 13.9 | $2.37 \pm 0.25$ | $7.34 \pm 0.02$ | $34.00 \pm 0.02$ | $5.66 \pm 0.04$ | $61.87 \pm 0.44$ |
| LinFullPost-MR | 16.8 | $\mathbf{1.82 \pm 0.15}$ | $7.35 \pm 0.02$ | $34.27 \pm 0.02$ | $7.85 \pm 0.07$ | $60.76 \pm 0.46$ |
| LinFullPost-SL | 18.1 | $2.62 \pm 0.27$ | $6.90 \pm 0.02$ | $33.91 \pm 0.02$ | $5.32 \pm 0.07$ | $60.89 \pm 0.47$ |
| ParamNoise | 27.4 | $2.77 \pm 0.15$ | $1.47 \pm 0.17$ | $26.81 \pm 0.10$ | $19.04 \pm 0.78$ | $68.92 \pm 0.53$ |
| ParamNoise-MR | 23 | $2.31 \pm 0.11$ | $1.76 \pm 0.18$ | $28.20 \pm 0.11$ | $20.25 \pm 0.41$ | $70.25 \pm 0.64$ |
| ParamNoise-SL | 20.9 | $2.49 \pm 0.09$ | $1.73 \pm 0.24$ | $25.63 \pm 0.09$ | $10.62 \pm 0.64$ | $66.75 \pm 0.54$ |
| Uniform | 51 | $100.00 \pm 0.15$ | $100.00 \pm 0.03$ | $100.00 \pm 0.01$ | $100.00 \pm 1.48$ | $100.00 \pm 1.01$ |

## 6 DISCUSSION

Overall, we found that there is significant room for improvement in uncertainty estimation for neural networks in sequential decision-making problems. First, unlike in supervised learning, sequential decision-making requires the model to be frequently updated as data is accumulated. As a result, methods that converge slowly are at a disadvantage because we must truncate optimization to make the method practical for the online setting. In these cases, we found that *partially* optimized uncertainty estimates can lead to catastrophic decisions and poor performance. Second, and while it deserves further investigation, it seems that *decoupling* representation learning and uncertainty estimation improves performance. The NeuralLinear algorithm is an example of this decoupling. With such a model, the uncertainty estimates can be solved for in closed form (but may be erroneous due to the simplistic model), so there is no issue with partial optimization. We suspect that this may be the reason for the improved performance. In addition, we observed that many algorithms are sensitive to their hyperparameters, so that best configurations are problem-dependent.

Finally, we found that in many cases, the inherit randomness in Stochastic Gradient Descent provided sufficient exploration. Accordingly, in some scenarios it may be hard to justify the use of complicated (and less transparent) variations of simple methods. However, Stochastic Gradient Descent is by no

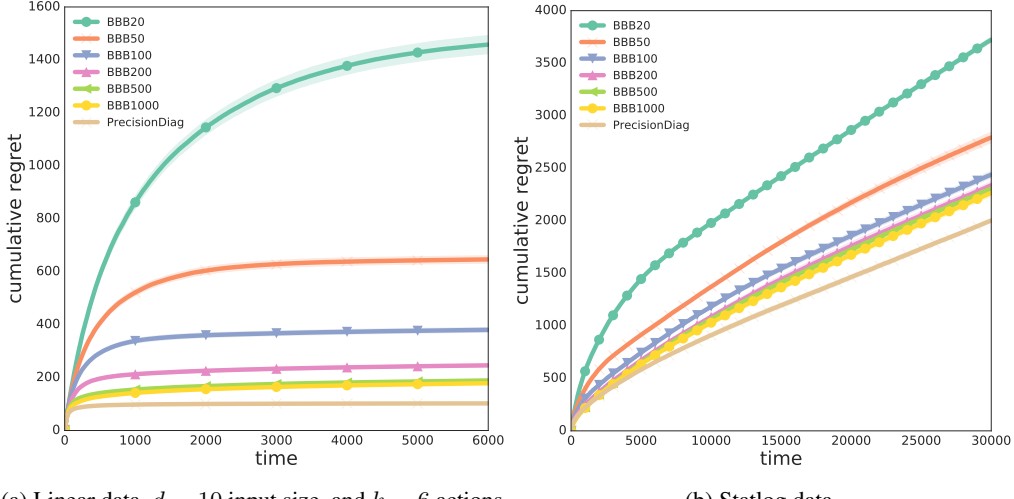

(a) Linear data, $d = 10$ input size, and $k = 6$ actions.  (b) Statlog data.

Figure 4: Cumulative regret for Bayes By Backprop (Variational Inference, fixed noise $\sigma = 0.75$) applied to a linear model and an exact mean field solution, denoted PrecisionDiag, with a linear bandit (left) and with the Statlog bandit (right). The suffix of the BBB legend label indicates the number of training epochs in each training step. We emphasize that in this evaluation, all algorithms use the same family of models (i.e., linear). While PrecisionDiag exactly solves the mean field problem, BBB relies on partial optimization via SGD. As the number of training epochs increases, BBB improves performance, but is always outperformed by PrecisionDiag.

means always enough: in our synthetic exploration-oriented problem (the Wheel bandit) additional exploration was necessary.

Next, we discuss our main findings for each class of algorithms.

**Linear Methods.** Linear methods offer a reasonable baseline, surprisingly strong in many cases. While their representation power is certainly a limiting factor, their ability to compute informative uncertainty measures seems to payoff and balance their initial disadvantage. They do well in several datasets, and are able to react fast to unexpected or extreme rewards (maybe as single points can have a heavy impact in fitted models, and their updates are immediate, deterministic, and exact). Some datasets clearly need more complex non-linear representations, and linear methods are unable to efficiently solve those. In addition, linear methods obviously offer computational advantages, and it would be interesting to investigate how their performance degrades when a finite data buffer feeds the estimates as various real-world online applications may require (instead of all collected data).

In terms of the diagonal linear approximations described in Section 3, we found that diagonalizing the precision matrix (as in mean-field Variational Inference) performs dramatically better than diagonalizing the covariance matrix.

**NeuralLinear.** The NeuralLinear algorithm sits near a sweet spot that is worth further studying. In general it seems to improve the RMS neural network it is based on, suggesting its exploration mechanisms add concrete value. We believe its main strength is that it is able to *simultaneously* learn a data representation that greatly simplifies the task at hand, and to accurately quantify the uncertainty over linear models that explain the observed rewards in terms of the proposed representation. While the former process may be noisier and heavily dependent on the amount of training steps that were taken and available data, the latter always offers the exact solution to its approximate parent problem. This, together with the partial success of linear methods with poor representations, may explain its promising results. In some sense, it knows what it knows. In the Wheel problem, which requires increasingly good exploration mechanisms, NeuralLinear is probably the best algorithm. Its performance is almost an order of magnitude better than any RMS algorithm (and its spinoffs, like Bootstrapped NN, Dropout, or Parameter Noise), and all greedy linear approaches. On the other hand, it is able to successfully solve problems that require non-linear representations (as Statlog or Covertype) where linear approaches fail. In addition, the algorithm is remarkably easy to tune, and

robust in terms of hyper-parameter configurations. While conceptually simple, its deployment to large scale systems may involve some technical difficulties; mainly, to update the Bayesian estimates when the network is re-trained. We believe, however, standard solutions to similar problems (like running averages) could greatly mitigate these issues. In our experiments and compared to other algorithms, as shown in Table 8, NeuralLinear is fast from a computational standpoint.

**Variational Inference.** Overall, Bayes By Backprop performed poorly, ranking in the bottom half of algorithms across datasets (Table 1). To investigate if this was due to underestimating uncertainty (as variational methods are known to (Bishop, 2006)), to the mean field approximation, or to stochastic optimization, we applied BBB to a linear model, where the mean field optimization problem can be solved in closed form (Figure 4). We found that the performance of BBB slowly improved as the number of training epochs increased, but underperformed compared to the exact mean field solution. Moreover, the difference in performance due to the number of training steps dwarfed the difference between the mean field solution and the exact posterior. This suggests that it is not sufficient to partially optimize the variational parameters when the uncertainty estimates directly affect the data being collected. In supervised learning, optimizing to convergence is acceptable, however in the online setting, optimizing to convergence at every step incurs unreasonable computational cost.

**Expectation-Propagation.** The performance of Black Box $\alpha$-divergence algorithms was poor. Because this class of algorithms is similar to BBB (in fact, as $\alpha \to 0$, it converges to the BBB objective), we suspect that partial convergence was also the cause of their poor performance. We found these algorithms to be sensitive to the number of training steps between actions, requiring a large number to achieve marginal performance. Their terrible performance in the Mushroom bandit is remarkable, while in the other datasets they perform slightly worse than their variational inference counterpart. Given the successes of Black Box $\alpha$-divergence in other domains (Hernández-Lobato et al., 2016), investigating approaches to sidestep the slow convergence of the uncertainty estimates is a promising direction for future work.

**Monte Carlo.** Constant-SGD comes out as the winner on Covertype, which requires non-linearity and exploration as evidenced by performance of the linear baseline approaches (Table 1). The method is especially appealing as it does not require tuning learning rates or exploration parameters. SGFS, however, performs better on average. The additional injected noise in SGFS may cause the model to explore more and thus perform better, as shown in the Wheel Bandit problem where SGFS strongly outperforms Constant-SGD.

**Bootstrap.** The bootstrap offers significant gains with respect to its parent algorithm (RMS) in several datasets. Note that in Statlog one of the actions is optimal around 80% of the time, and the bootstrapped predictions may help to avoid getting stuck, something from which RMS methods may suffer. In other scenarios, the randomness from SGD may be enough for exploration, and the bootstrap may not offer important benefits. In those cases, it might not justify the heavy computational overhead of the method. We found it surprising that the optimized versions of BootstrappedNN decided to use *only* $q = 2$ and $q = 3$ networks respectively (while we set its value to $q = 10$ in the manually tuned version, and the extra networks did not improve performance significantly). Unfortunately, Bootstrapped NNs were not able to solve the Wheel problem, and its performance was fairly similar to that of RMS. One possible explanation is that —given the sparsity of the reward— all the bootstrapped networks agreed for the most part, and the algorithm simply got stuck selecting action $a_1$. As opposed to linear models, reacting to unusual rewards could take Bootstrapped NNs some time as good predictions could be randomly overlooked (and useful data discarded if $p \ll 1$).

**Direct Noise Injection.** When properly tuned, Parameter-Noise provided an important boost in performance across datasets over the learner that it was based on (RMS), average rank of ParamNoise-SL is 20.9 compared to RMS at 28.7 (Table 1). However, we found the algorithm hard to tune and sensitive to the heuristic controlling the injected noise-level. On the synthetic Wheel problem —where exploration is necessary— both parameter-noise and RMS suffer from underexploration and perform similarly, except ParamNoise-MR which does a good job. In addition, developing an intuition for the heuristic is not straightforward as it lacks transparency and a principled grounding, and thus may require repeated access to the decision-making process for tuning.

**Dropout.** We initially experimented with two dropout versions: fixed $p = 0.5$, and $p = 0.8$. The latter consistently delivered better results, and it is the one we manually picked. The optimized versions of the algorithm provided decent improvements over its base RMS (specially Dropout-MR).

In the Wheel problem, dropout performance is somewhat poor: Dropout is outperformed by RMS, while Dropout-MR offers gains with respect to all versions of RMS but it is not competitive with the best algorithms. Overall, the algorithm seems to heavily depend on its hyper-parameters (see cum-regret performance of the raw Dropout, for example). Dropout was used both for training and for decision-making; unfortunately, we did not add a baseline where dropout only applies during training. Consequently, it is not obvious how to disentangle the contribution of better training from that of better exploration. This remains as future work.

**Bayesian Non-parametrics.** Perhaps unsurprisingly, Gaussian processes perform reasonably well on problems with little data but struggle on larger problems. While this motivated the use of sparse GP, the latter was not able to perform similarly to stronger (and definitively simpler) methods.

# 7 Conclusions and Future Work

In this work, we empirically studied the impact on performance of approximate model posteriors for decision making via Thompson Sampling in contextual bandits. We found that the most robust methods exactly measured uncertainty (possibly under the wrong model assumptions) on top of complex representations learned in parallel. More complicated approaches that learn the representation and its uncertainty together seemed to require heavier training, an important drawback in online scenarios, and exhibited stronger hyper-parameter dependence. Further exploring and developing the promising approaches is an exciting avenue for future work.

## Acknowledgments

We are extremely grateful to Dan Moldovan, Sven Schmit, Matt Hoffman, Matt Johnson, Ramon Iglesias, and Rif Saurous for their valuable feedback and comments. We also thank the anonymous reviewers, whose suggestions truly helped improve the current work.

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

APPENDIX

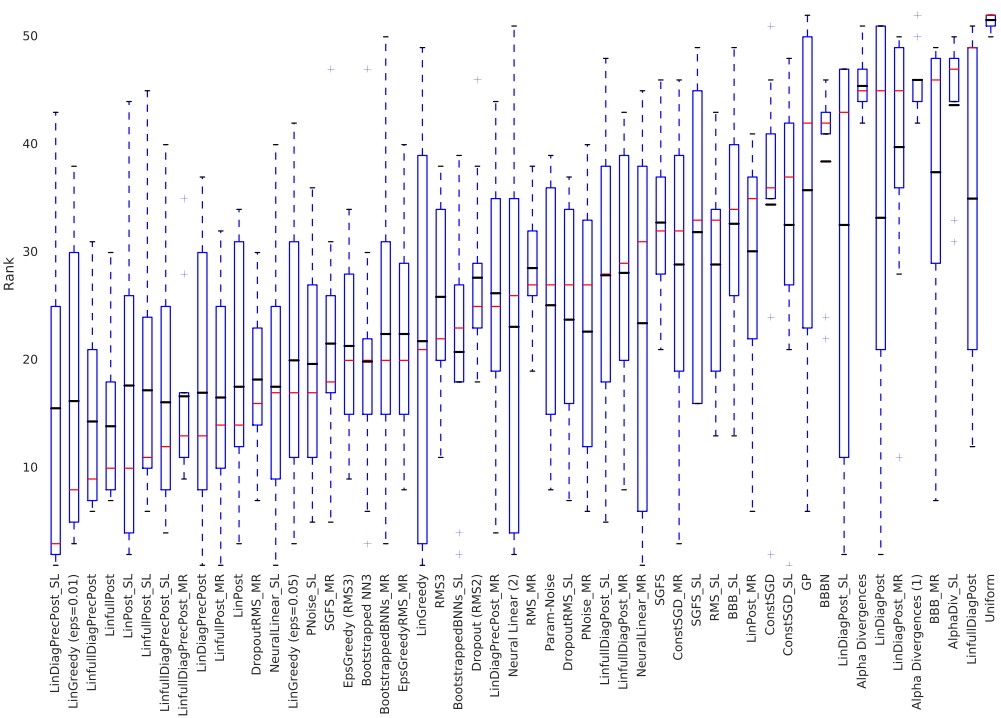

Figure 5: A boxplot of the ranks achieved by each algorithm across the suite of benchmarks. The red and black solid lines respectively indicate the median and mean rank across problems.

Table 2: Detailed description of the algorithms in the experiments. Unless otherwise stated, algorithms use $t_s = 20$ (mini-batches per training period), and $t_f = 20$ (one training period every $t_f$ contexts).

| Algorithm | Description |
|---|---|
| AlphaDivergence (1) | BB $\alpha$-divergence with $\alpha = 0.5$, noise $\sigma = 0.1$, $K = 10$, prior var $\sigma_0^2 = 0.1$. ($t_s = 100$, first 100 times linear decay from $t_s = 10000$). |
| AlphaDivergence | BB $\alpha$-divergence with $\alpha = 0.1$, noise $\sigma = 0.1$, $K = 10$, prior var $\sigma_0^2 = 0.1$. ($t_s = 100$, first 100 times linear decay from $t_s = 10000$). |
| AlphaDivergence-SL | BB $\alpha$-divergence with $\alpha = 0.01$, noise $\sigma = 2.79$, $K = 1$, prior var $\sigma_0^2 = 0.18$. ($t_s = 200$, first 100 times linear decay from $t_s = 10000$). |
| BBB | BayesByBackprop with noise $\sigma = 0.1$. ($t_s = 100$, first 100 times linear decay from $t_s = 10000$). |
| BBB-MR | BayesByBackprop with noise $\sigma = 1.3$, and prior $\sigma_p = 1.48$. ($t_s = 50$, first 100 times linear decay from $t_s = 10000$). |
| BBB-SL | BayesByBackprop with noise $\sigma = 0.03$, and prior $\sigma_p = 2.86$. ($t_s = 100$, first 100 times linear decay from $t_s = 10000$). |
| BootstrappedNN | Bootstrapped with $q = 10$ models, and $p = 1.0$. Based on RMS3 net. |
| BootstrappedNN-MR | Bootstrapped with $q = 2$ models, $p = 0.95$, $t_s = 50$, $t_f = 50$. Based on RMS2 net. |
| BootstrappedNN-SL | Bootstrapped with $q = 3$ models, $p = 0.92$, $t_s = 20$, $t_f = 20$. Based on RMS3 net. |
| Dropout | Dropout with probability $p = 0.8$. Based on RMS2 net. |
| Dropout-MR | Dropout with probability $p = 0.8$, $t_s = 50$, $t_f = 50$. Based on RMS2 net. |
| Dropout-SL | Dropout with probability $p = 0.95$, $t_s = 100$, $t_f = 5$. Based on RMS3 net. |
| GP | For computational reasons, it only uses the first 1000 data points. |
| NeuralLinear | Noise prior $a_0 = 3$, $b_0 = 3$. Ridge prior $\lambda = 0.25$. Based on RMS2 net. Trained for $t_s = 20$, $t_f = 20$. |
| NeuralLinear-MR | Noise prior $a_0 = 12$, $b_0 = 30$. Ridge prior $\lambda = 23.0$. Based on RMS2 net. Trained for $t_s = 50$, $t_f = 20$. |
| NeuralLinear-SL | Noise prior $a_0 = 38$, $b_0 = 1$. Ridge prior $\lambda = 1.5$. Based on RMS2 net. Trained for $t_s = 20$, $t_f = 20$. |
| RMS1 | Greedy NN approach, fixed learning rate ($\gamma = 0.01$). |
| RMS2 | Learning rate decays, and it is reset every training period. |
| RMS3 | Learning rate decays, and it is not reset at all. It starts at $\gamma = 1$. |
| RMS | Based on RMS3 net. Learning decay rate is $0.55$, initial learning rate is $1.0$. Trained for $t_s = 100$, $t_f = 20$. |
| RMS-MR | Based on RMS3 net. Learning decay rate is $2.5$, initial learning rate is $1.0$. Trained for $t_s = 50$, $t_f = 20$. |
| RMS-SL | Based on RMS3 net. Learning decay rate is $0.4$, initial learning rate is $1.1$. Trained for $t_s = 100$, $t_f = 20$. |
| SGFS | Burning $= 500$, learning rate $\gamma = 0.014$, EMA decay $= 0.9$, noise $\sigma = 0.75$. |
| SGFS-MR | Burning $= 100$, learning rate $\gamma = 0.19$, EMA decay $= 0.23$, noise $\sigma = 0.33$. |
| SGFS-SL | Burning $= 2000$, learning rate $\gamma = 0.15$, EMA decay $= 0.58$, noise $\sigma = 0.34$. |
| ConstSGD | Burning $= 500$, EMA decay $= 0.9$, noise $\sigma = 0.5$. |
| ConstSGD-MR | Burning $= 50$, EMA decay $= 0.87$, noise $\sigma = 0.44$. Trained for $t_s = 20$, $t_f = 50$. |
| ConstSGD-SL | Burning $= 500$, EMA decay $= 0.82$, noise $\sigma = 1.05$. Trained for $t_s = 20$, $t_f = 10$. |
| EpsGreedyRMS | Initial $\epsilon = 0.01$, multiplied by $0.999$ after every context. Based on RMS3 net. |
| EpsGreedyRMS-MR | Initial $\epsilon = 0.046$, multiplied by $0.93$ after every context. Based on RMS3 net. Trained for $t_s = 50$, $t_f = 20$. |
| EpsGreedyRMS-SL | Initial $\epsilon = 0.23$, multiplied by $0.95$ after every context. Based on RMS2 net. Trained for $t_s = 50$, $t_f = 10$. |
| LinPost | Ridge prior $\lambda = 0.25$. Assumed noise level $\sigma^2 = 0.25$. |
| LinPost-MR | Ridge prior $\lambda = 11.12$. Assumed noise level $\sigma^2 = 2.0$. |
| LinPost-SL | Ridge prior $\lambda = 37.58$. Assumed noise level $\sigma^2 = 0.037$. |
| LinDiagPost | $\Sigma$ in Eq. 1 is diagonalized. Ridge prior $\lambda = 0.25$. Assumed noise level $\sigma^2 = 0.25$. |
| LinDiagPost-MR | $\Sigma$ in Eq. 1 is diagonalized. Ridge prior $\lambda = 14.20$. Assumed noise level $\sigma^2 = 2.49$. |
| LinDiagPost-SL | $\Sigma$ in Eq. 1 is diagonalized. Ridge prior $\lambda = 40.0$. Assumed noise level $\sigma^2 = 0.011$. |
| LinDiagPrecPost | $\Sigma^{-1}$ in Eq. 1 is diagonalized. Ridge prior $\lambda = 0.25$. Assumed noise level $\sigma^2 = 0.25$. |
| LinDiagPrecPost-MR | $\Sigma^{-1}$ in Eq. 1 is diagonalized. Ridge prior $\lambda = 37.35$. Assumed noise level $\sigma^2 = 0.68$. |
| LinDiagPrecPost-SL | $\Sigma^{-1}$ in Eq. 1 is diagonalized. Ridge prior $\lambda = 13.49$. Assumed noise level $\sigma^2 = 0.01$. |
| LinGreedy | Takes action with highest predicted reward for Ridge regression, $\lambda = 0.25$. Noise level $\sigma^2 = 0.25$. |
| LinGreedy (eps = 0.01) | linGreedy that selects action uniformly at random with prob $p = 0.01$. |
| LinGreedy (eps = 0.05) | linGreedy that selects action uniformly at random with prob $p = 0.05$. |
| LinFullPost | Noise prior $a_0 = 6$, $b_0 = 6$. Ridge prior $\lambda = 0.25$. |
| LinFullPost-MR | Noise prior $a_0 = 30.0$, $b_0 = 35.0$. Ridge prior $\lambda = 20.0$. |
| LinFullPost-SL | Noise prior $a_0 = 35.0$, $b_0 = 5.0$. Ridge prior $\lambda = 20.0$. |
| LinFullDiagPost | $\Sigma$ in Eq. 1 is diagonalized. Noise prior $a_0 = 6$, $b_0 = 6$. Ridge prior $\lambda = 0.25$. |
| LinFullDiagPost-MR | $\Sigma$ in Eq. 1 is diagonalized. Noise prior $a_0 = 22.27$, $b_0 = 35.89$. Ridge prior $\lambda = 39.95$. |
| LinFullDiagPost-SL | $\Sigma$ in Eq. 1 is diagonalized. Noise prior $a_0 = 39.94$, $b_0 = 0.03$. Ridge prior $\lambda = 39.74$. |
| LinFullDiagPrecPost | $\Sigma$ in Eq. 1 is diagonalized. Noise prior $a_0 = 6$, $b_0 = 6$. Ridge prior $\lambda = 0.25$. |
| LinFullDiagPrecPost-MR | $\Sigma$ in Eq. 1 is diagonalized. Noise prior $a_0 = 0.23$, $b_0 = 21.23$. Ridge prior $\lambda = 2.21$. |
| LinFullDiagPrecPost-SL | $\Sigma$ in Eq. 1 is diagonalized. Noise prior $a_0 = 4.25$, $b_0 = 0.073$. Ridge prior $\lambda = 13.07$. |
| ParamNoise | Layer normalization. Initial noise $\sigma = 0.01$, and level $\epsilon = 0.01$. Based on RMS2 net. |
| ParamNoise-MR | Layer normalization. Initial noise $\sigma = 2.6$, and level $\epsilon = 1.5$. Based on RMS3 net, $t_s = 20$, $t_f = 20$. |
| ParamNoise-SL | Layer normalization. Initial noise $\sigma = 1.8$, and level $\epsilon = 2.0$. Based on RMS2 net, $t_s = 20$, $t_f = 50$. |
| Uniform | Takes each action at random with equal probability. |

Table 3: Detailed description of the algorithms in the *linear* experiments. Unless otherwise stated, algorithms use $t_s = 100$ (mini-batches per training period), and $t_f = 20$ (one training period every $t_f$ contexts).

| Algorithm | Description |
|---|---|
| Alpha Divergences | BB $\alpha$-divergence with $\alpha = 0.1$, noise $\sigma = 0.1$, $K = 10$, prior var $\sigma_0^2 = 0.1$. ($t_s = 100$, first 100 times linear decay from $t_s = 10000$). |
| Alpha Divergences (1) | BB $\alpha$-divergence with $\alpha = 0.5$, noise $\sigma = 0.1$, $K = 10$, prior var $\sigma_0^2 = 0.1$. ($t_s = 100$, first 100 times linear decay from $t_s = 10000$). |
| Alpha Divergences (2) | BB $\alpha$-divergence with $\alpha = 1.0$, noise $\sigma = 0.1$, $K = 10$, prior var $\sigma_0^2 = 0.1$. ($t_s = 100$, first 100 times linear decay from $t_s = 10000$). |
| Alpha Divergences (3) | BB $\alpha$-divergence with $\alpha = 0.5$, noise $\sigma = 0.1$, $K = 10$, prior var $\sigma_0^2 = 1.0$. ($t_s = 100$, first 100 times linear decay from $t_s = 10000$). |
| BBBN | BayesByBackprop with noise $\sigma = 0.1$. ($t_s = 100$, first 100 times linear decay from $t_s = 10000$). |
| BBBN2 | BayesByBackprop with noise $\sigma = 0.5$. ($t_s = 100$, first 100 times linear decay from $t_s = 10000$). |
| BBBN3 | BayesByBackprop with noise $\sigma = 0.75$. ($t_s = 100$, first 100 times linear decay from $t_s = 10000$). |
| BBBN4 | BayesByBackprop with noise $\sigma = 1.0$. ($t_s = 100$, first 100 times linear decay from $t_s = 10000$). |
| Bootstrapped NN | Bootstrapped with $q = 5$ models, and $p = 0.85$. Based on RMS3 net. |
| Bootstrapped NN2 | Bootstrapped with $q = 5$ models, and $p = 1.0$. Based on RMS3 net. |
| Bootstrapped NN3 | Bootstrapped with $q = 10$ models, and $p = 1.0$. Based on RMS3 net. |
| Dropout (RMS3) | Dropout with probability $p = 0.8$. Based on RMS3 net. |
| Dropout (RMS2) | Dropout with probability $p = 0.8$. Based on RMS2 net. |
| RMS1 | Greedy NN approach, fixed learning rate ($\gamma = 0.01$). |
| RMS2 | Learning rate decays, and it is reset every training period. |
| RMS2b | Similar to RMS2, but training for longer ($t_s = 800$). |
| RMS3 | Learning rate decays, and it is not reset at all. Starts at $\gamma = 1$. |
| SGFS | Burning $= 500$, learning rate $\gamma = 0.014$, EMA decay $= 0.9$, noise $\sigma = 0.75$. |
| ConstSGD | Burning $= 500$, EMA decay $= 0.9$, noise $\sigma = 0.5$. |
| EpsGreedy (RMS1) | Initial $\epsilon = 0.01$. Multiplied by 0.999 after every context. Based on RMS1 net. |
| EpsGreedy (RMS2) | Initial $\epsilon = 0.01$. Multiplied by 0.999 after every context. Based on RMS2 net. |
| EpsGreedy (RMS3) | Initial $\epsilon = 0.01$. Multiplied by 0.999 after every context. Based on RMS3 net. |
| LinDiagPost | $\Sigma$ in Eq. 1 is diagonalized. Ridge prior $\lambda = 0.25$. Assumed noise level $\sigma^2 = 0.25$. |
| LinDiagPrecPost | $\Sigma^{-1}$ in Eq. 1 is diagonalized. Ridge prior $\lambda = 0.25$. Assumed noise level $\sigma^2 = 0.25$. |
| LinGreedy | Takes action with highest predicted reward for Ridge regression, $\lambda = 0.25$. Noise level $\sigma^2 = 0.25$. |
| LinGreedy (eps = 0.01) | linGreedy that selects action uniformly at random with prob $p = 0.01$. |
| LinGreedy (eps = 0.05) | linGreedy that selects action uniformly at random with prob $p = 0.05$. |
| LinPost | Ridge prior $\lambda = 0.25$. Assumed noise level $\sigma^2 = 0.25$. |
| LinFullDiagPost | $\Sigma$ in Eq. 1 is diagonalized. Noise prior $a_0 = 6$, $b_0 = 6$. Ridge prior $\lambda = 0.25$. |
| LinFullDiagPrecPost | $\Sigma^{-1}$ in Eq. 1 is diagonalized. Noise prior $a_0 = 6$, $b_0 = 6$. Ridge prior $\lambda = 0.25$. |
| LinFullPost | Noise prior $a_0 = 6$, $b_0 = 6$. Ridge prior $\lambda = 0.25$. |
| Param-Noise | Initial noise $\sigma = 0.01$, and level $\epsilon = 0.01$. Based on RMS3 net. |
| Param-Noise2 | Initial noise $\sigma = 0.01$, and level $\epsilon = 0.01$. Based on RMS3 net. Trained for longer: $t_s = 800$. |
| Uniform | Takes each action at random with equal probability. |

Table 4: Cumulative regret incurred by *linear* models using algorithms in Section 3 on the bandits described in Section A. Values reported are the mean over 50 independent trials with standard error of the mean.

| | Mushroom | Statlog | Covertype | Financial | Jester | Adult |
|---|---|---|---|---|---|---|
| **Cumulative regret** | | | | | | |
| Alpha Divergences | 29051.02 ± 238.56 | 3213.57 ± 8.88 | 47594.06 ± 30.86 | 119.92 ± 1.70 | 60401.66 ± 455.68 | 34774.94 ± 18.70 |
| Alpha Divergences (1) | 65203.06 ± 204.46 | 3636.06 ± 19.61 | 47904.47 ± 27.89 | 118.67 ± 1.86 | 61896.47 ± 380.77 | 35043.71 ± 15.02 |
| Alpha Divergences (2) | 67612.86 ± 195.67 | 5644.49 ± 24.17 | 50401.92 ± 35.04 | 131.40 ± 2.05 | 69341.27 ± 587.65 | 35925.67 ± 12.82 |
| Alpha Divergences (3) | 77061.53 ± 221.74 | 3670.65 ± 17.56 | 47973.43 ± 35.15 | 164.44 ± 2.89 | 62070.93 ± 452.55 | 35193.04 ± 16.59 |
| BBBN | 19352.55 ± 2315.29 | **2611.59 ± 9.19** | **43684.69 ± 59.12** | 132.12 ± 11.53 | **56384.55 ± 477.64** | **31650.61 ± 15.92** |
| BBBN2 | 24885.00 ± 4784.46 | 2820.29 ± 7.58 | 44603.71 ± 21.78 | 232.86 ± 2.07 | **56465.72 ± 498.79** | 33540.20 ± 15.02 |
| BBBN3 | 43362.04 ± 7452.98 | 3065.43 ± 11.70 | 45880.12 ± 21.76 | 361.13 ± 2.62 | 57055.81 ± 492.60 | 35011.00 ± 16.26 |
| BBBN4 | 61581.02 ± 8644.66 | 3365.49 ± 15.22 | 47324.12 ± 21.01 | 496.80 ± 3.84 | **55826.75 ± 525.71** | 36290.27 ± 16.80 |
| Bootstrapped NN | 15207.55 ± 2223.28 | 2801.43 ± 58.40 | 48399.53 ± 276.16 | 149.15 ± 12.35 | 57922.32 ± 470.78 | 31740.18 ± 35.07 |
| Bootstrapped NN2 | 30388.98 ± 3002.15 | 2777.61 ± 85.16 | 49919.61 ± 421.32 | 124.47 ± 8.93 | 56886.28 ± 522.05 | 31760.43 ± 53.26 |
| Bootstrapped NN3 | 22681.53 ± 2445.45 | **2574.35 ± 38.98** | | **108.00 ± 6.96** | 56693.99 ± 497.70 | 31845.12 ± 63.36 |
| Dropout (RMS3) | 24079.80 ± 2905.35 | 3504.33 ± 233.49 | 49382.00 ± 336.47 | **118.64 ± 6.63** | **56600.80 ± 562.76** | 32133.98 ± 67.08 |
| Dropout (RMS2) | 30470.00 ± 3256.06 | 3594.18 ± 224.22 | 49555.14 ± 354.48 | 127.23 ± 8.85 | 57321.68 ± 496.09 | 32132.51 ± 81.59 |
| RMS1 | 19626.63 ± 2706.22 | 3147.37 ± 12.28 | 49518.84 ± 234.92 | 126.73 ± 9.38 | 58098.59 ± 546.44 | 33514.08 ± 20.16 |
| RMS2 | 24920.71 ± 3629.18 | 3511.47 ± 237.12 | 49475.98 ± 355.82 | 129.84 ± 9.76 | **55619.60 ± 476.35** | 32183.02 ± 71.02 |
| RMS2b | 35375.51 ± 3941.93 | 4128.22 ± 273.86 | 49554.74 ± 461.20 | 128.02 ± 10.80 | **55941.69 ± 554.92** | 32094.82 ± 78.15 |
| RMS3 | 24408.57 ± 2876.77 | 3288.41 ± 198.61 | 50056.27 ± 392.93 | **109.53 ± 7.92** | 56661.44 ± 495.01 | 32087.24 ± 65.81 |
| SGFS | 129586.43 ± 82.79 | 10811.00 ± 186.70 | 79162.47 ± 375.83 | 4747.22 ± 64.59 | 67961.22 ± 1393.66 | 40784.20 ± 87.79 |
| ConstSGD | 129737.86 ± 85.36 | 9405.92 ± 0.33 | 79492.84 ± 927.87 | 4749.99 ± 72.03 | 65572.88 ± 1112.01 | 39450.41 ± 33.53 |
| EpsGreedy (RMS1) | 8732.76 ± 516.32 | 3283.57 ± 11.93 | 48605.16 ± 125.63 | 143.89 ± 4.31 | 56961.95 ± 444.26 | 33569.51 ± 19.84 |
| EpsGreedy (RMS2) | 23687.76 ± 2166.78 | 3009.10 ± 124.54 | 48617.06 ± 331.37 | 146.49 ± 5.56 | **55855.45 ± 457.55** | 32021.98 ± 66.44 |
| EpsGreedy (RMS3) | 23637.96 ± 1847.91 | 2938.84 ± 80.86 | 48509.29 ± 273.31 | 145.17 ± 5.62 | **56072.51 ± 479.59** | 32026.18 ± 67.21 |
| LinDiagPost | 42696.84 ± 276.34 | 19118.47 ± 12.30 | 122826.24 ± 24.59 | 459.02 ± 2.95 | **56568.08 ± 527.28** | 41858.82 ± 9.56 |
| LinDiagPrecPost | 19523.67 ± 2404.15 | 2805.69 ± 9.76 | 44253.76 ± 50.98 | 217.08 ± 2.17 | **56227.81 ± 463.11** | 33521.02 ± 15.75 |
| LinGreedy | 26835.20 ± 3759.77 | 3897.45 ± 215.39 | 45446.98 ± 166.07 | **105.48 ± 6.73** | **55651.56 ± 438.66** | 34558.63 ± 315.75 |
| LinGreedy (eps=0.01) | 8522.86 ± 537.80 | 3775.86 ± 143.74 | 44407.24 ± 81.43 | 129.20 ± 3.84 | 57082.85 ± 538.96 | 32214.82 ± 114.46 |
| LinGreedy (eps=0.05) | 14489.69 ± 145.06 | 4814.65 ± 66.76 | 47485.59 ± 42.17 | 308.34 ± 4.74 | 58201.11 ± 460.96 | 31813.22 ± 19.33 |
| LinPost | 16251.22 ± 1680.93 | 2849.12 ± 6.62 | 44176.49 ± 22.51 | 347.11 ± 2.69 | 56677.18 ± 550.34 | 33243.59 ± 16.12 |
| LinfullDiagPost | 212029.08 ± 387.36 | 10546.86 ± 9.82 | 94875.53 ± 34.06 | 336.08 ± 2.70 | 59981.66 ± 388.85 | 40372.00 ± 12.03 |
| LinfullDiagPrecPost | 10485.82 ± 1081.24 | 2738.04 ± 7.74 | **43734.08 ± 26.14** | 189.89 ± 2.13 | 57738.18 ± 428.36 | 32329.84 ± 12.74 |
| LinfullPost | **5299.90 ± 532.93** | 2746.51 ± 6.46 | **43728.67 ± 25.12** | 267.23 ± 2.33 | 59037.26 ± 463.65 | 32233.51 ± 14.20 |
| Param-Noise | 27804.49 ± 3389.40 | 2888.02 ± 61.41 | 49353.51 ± 364.93 | 142.05 ± 13.22 | 57399.91 ± 543.27 | 32091.20 ± 80.79 |
| Param-Noise2 | 22882.35 ± 2682.16 | 2853.94 ± 57.96 | 49426.47 ± 326.31 | **107.94 ± 5.45** | 57418.49 ± 530.30 | 32090.24 ± 66.27 |
| Uniform | 244892.76 ± 412.03 | 37288.98 ± 8.71 | 128543.61 ± 18.59 | 4672.92 ± 57.66 | 95646.52 ± 1145.95 | 41989.37 ± 8.38 |

Table 5: Simple regret incurred by *linear* models using algorithms in Section 3 on the bandits described in Section A. Simple regret was approximated by averaging the regret over the final 500 steps. Values reported are the mean over 50 independent trials with standard error of the mean.

| | Mushroom | Statlog | Covertype | Financial | Jester | Adult |
|---|---|---|---|---|---|---|
| **Simple regret** | | | | | | |
| Alpha Divergences | 0.68 ± 0.04 | 0.07 ± 0.00 | 0.31 ± 0.00 | **0.00 ± 0.00** | 2.91 ± 0.04 | 0.75 ± 0.00 |
| Alpha Divergences (1) | 1.50 ± 0.05 | 0.08 ± 0.00 | 0.31 ± 0.00 | **0.00 ± 0.00** | 2.98 ± 0.03 | 0.75 ± 0.00 |
| Alpha Divergences (2) | 1.51 ± 0.05 | 0.13 ± 0.00 | 0.32 ± 0.00 | 0.00 ± 0.00 | 3.42 ± 0.05 | 0.77 ± 0.00 |
| Alpha Divergences (3) | 1.50 ± 0.05 | 0.08 ± 0.00 | 0.31 ± 0.00 | **0.00 ± 0.00** | 2.97 ± 0.03 | 0.75 ± 0.00 |
| BBBN | 0.35 ± 0.05 | 0.06 ± 0.00 | **0.28 ± 0.00** | 0.01 ± 0.00 | **2.78 ± 0.04** | **0.67 ± 0.00** |
| BBBN2 | 0.45 ± 0.09 | 0.06 ± 0.00 | **0.28 ± 0.00** | 0.01 ± 0.00 | **2.77 ± 0.04** | 0.70 ± 0.00 |
| BBBN3 | 0.81 ± 0.15 | 0.06 ± 0.00 | 0.29 ± 0.00 | 0.03 ± 0.00 | **2.78 ± 0.04** | 0.72 ± 0.00 |
| BBBN4 | 1.13 ± 0.18 | 0.06 ± 0.00 | 0.29 ± 0.00 | 0.05 ± 0.00 | **2.77 ± 0.04** | 0.75 ± 0.00 |
| Bootstrapped NN | 0.22 ± 0.04 | **0.05 ± 0.00** | 0.31 ± 0.00 | 0.01 ± 0.00 | 2.82 ± 0.04 | **0.68 ± 0.00** |
| Bootstrapped NN2 | 0.51 ± 0.06 | **0.05 ± 0.00** | 0.33 ± 0.00 | 0.01 ± 0.00 | 2.76 ± 0.03 | **0.68 ± 0.00** |
| Bootstrapped NN3 | 0.36 ± 0.05 | **0.05 ± 0.00** | | 0.01 ± 0.00 | 2.82 ± 0.04 | 0.68 ± 0.00 |
| Dropout (RMS3) | 0.41 ± 0.06 | 0.07 ± 0.01 | 0.32 ± 0.00 | 0.01 ± 0.00 | **2.78 ± 0.03** | 0.69 ± 0.00 |
| Dropout (RMS2) | 0.56 ± 0.06 | 0.07 ± 0.01 | 0.32 ± 0.00 | 0.01 ± 0.00 | 2.81 ± 0.03 | 0.68 ± 0.00 |
| RMS1 | 0.33 ± 0.05 | 0.07 ± 0.00 | 0.32 ± 0.00 | 0.01 ± 0.00 | 2.90 ± 0.04 | 0.73 ± 0.00 |
| RMS2 | 0.42 ± 0.07 | 0.07 ± 0.01 | 0.33 ± 0.00 | 0.01 ± 0.00 | **2.72 ± 0.03** | 0.69 ± 0.00 |
| RMS2b | 0.61 ± 0.08 | 0.08 ± 0.01 | 0.32 ± 0.01 | 0.01 ± 0.00 | 2.75 ± 0.03 | 0.69 ± 0.00 |
| RMS3 | 0.39 ± 0.05 | 0.06 ± 0.00 | 0.33 ± 0.00 | 0.01 ± 0.00 | 2.84 ± 0.04 | 0.68 ± 0.00 |
| SGFS | 2.59 ± 0.02 | 0.21 ± 0.00 | 0.53 ± 0.01 | 1.28 ± 0.02 | 3.55 ± 0.08 | 0.90 ± 0.01 |
| ConstSGD | 2.58 ± 0.01 | 0.21 ± 0.00 | 0.53 ± 0.01 | 1.28 ± 0.02 | 3.43 ± 0.07 | 0.87 ± 0.00 |
| EpsGreedy (RMS1) | **0.08 ± 0.01** | 0.07 ± 0.00 | 0.32 ± 0.00 | 0.01 ± 0.00 | 2.80 ± 0.04 | 0.73 ± 0.00 |
| EpsGreedy (RMS2) | 0.32 ± 0.04 | 0.05 ± 0.00 | 0.32 ± 0.00 | 0.01 ± 0.00 | **2.78 ± 0.03** | 0.69 ± 0.00 |
| EpsGreedy (RMS3) | 0.32 ± 0.03 | **0.05 ± 0.00** | 0.32 ± 0.00 | 0.02 ± 0.00 | **2.78 ± 0.04** | 0.68 ± 0.00 |
| LinDiagPost | 0.80 ± 0.02 | 0.32 ± 0.00 | 0.82 ± 0.00 | 0.03 ± 0.00 | 2.75 ± 0.04 | 0.92 ± 0.00 |
| LinDiagPrecPost | 0.37 ± 0.05 | 0.05 ± 0.00 | **0.28 ± 0.00** | 0.01 ± 0.00 | **2.77 ± 0.04** | 0.69 ± 0.00 |
| LinGreedy | 0.51 ± 0.08 | 0.07 ± 0.01 | 0.30 ± 0.00 | 0.01 ± 0.00 | **2.73 ± 0.03** | 0.74 ± 0.01 |
| LinGreedy (eps=0.01) | **0.07 ± 0.01** | 0.07 ± 0.00 | 0.29 ± 0.00 | 0.01 ± 0.00 | 2.80 ± 0.03 | **0.67 ± 0.00** |
| LinGreedy (eps=0.05) | 0.24 ± 0.02 | 0.10 ± 0.00 | 0.31 ± 0.00 | 0.06 ± 0.00 | 2.86 ± 0.03 | 0.68 ± 0.00 |
| LinPost | 0.29 ± 0.03 | 0.06 ± 0.00 | **0.28 ± 0.00** | 0.01 ± 0.00 | **2.74 ± 0.04** | 0.69 ± 0.00 |
| LinfullDiagPost | 4.10 ± 0.07 | 0.18 ± 0.00 | 0.63 ± 0.00 | 0.00 ± 0.00 | 2.86 ± 0.03 | 0.89 ± 0.00 |
| LinfullDiagPrecPost | 0.19 ± 0.02 | 0.05 ± 0.00 | **0.28 ± 0.00** | 0.00 ± 0.00 | 2.82 ± 0.03 | **0.67 ± 0.00** |
| LinfullPost | **0.08 ± 0.01** | 0.05 ± 0.00 | **0.28 ± 0.00** | 0.00 ± 0.00 | 2.86 ± 0.03 | **0.67 ± 0.00** |
| Param-Noise | 0.49 ± 0.07 | **0.05 ± 0.00** | 0.32 ± 0.00 | 0.01 ± 0.00 | 2.87 ± 0.04 | 0.69 ± 0.00 |
| Param-Noise2 | 0.36 ± 0.05 | **0.05 ± 0.00** | 0.33 ± 0.00 | 0.01 ± 0.00 | 2.83 ± 0.04 | 0.69 ± 0.00 |
| Uniform | 4.88 ± 0.07 | 0.86 ± 0.00 | 0.86 ± 0.00 | 1.25 ± 0.02 | 5.03 ± 0.07 | 0.93 ± 0.00 |

Table 6: Cumulative regret incurred by models using algorithms in Section 3 on the bandits described in Section A. Values reported are the mean over 50 independent trials with standard error of the mean. Normalized with respect to the performance of Uniform.

| | Mushroom | Statlog | Covertype | Financial | Jester | Adult | Song | Census |
|---|---|---|---|---|---|---|---|---|
| AlphaDivergence (1) | 54.29 ± 0.04 | 19.35 ± 1.72 | 39.42 ± 0.50 | 40.10 ± 0.69 | 72.99 ± 0.54 | 94.34 ± 0.03 | 97.65 ± 0.23 | 67.23 ± 0.37 |
| AlphaDivergence | 54.17 ± 0.03 | 19.30 ± 0.84 | 44.31 ± 0.77 | 47.76 ± 0.89 | 71.86 ± 0.72 | 94.13 ± 0.03 | 96.99 ± 0.20 | 64.26 ± 0.63 |
| AlphaDivergence-SL | 53.88 ± 0.03 | 21.12 ± 1.14 | 60.05 ± 0.02 | 70.19 ± 3.50 | 69.11 ± 0.75 | 94.13 ± 0.03 | 99.42 ± 0.05 | 67.84 ± 0.06 |
| BBB | 3.57 ± 0.20 | 12.61 ± 1.53 | 58.19 ± 2.16 | 22.55 ± 1.27 | 71.43 ± 0.67 | 94.03 ± 0.59 | 97.35 ± 0.37 | 65.99 ± 2.74 |
| BBB-MR | 10.93 ± 2.69 | 25.29 ± 0.00 | 60.92 ± 0.55 | 59.84 ± 2.89 | 65.01 ± 0.74 | 94.61 ± 0.36 | 95.77 ± 0.24 | 68.57 ± 1.01 |
| BBB-SL | 4.08 ± 0.19 | 1.86 ± 0.29 | 38.50 ± 0.97 | 13.76 ± 0.60 | 74.70 ± 0.68 | 90.28 ± 0.68 | 96.13 ± 0.26 | 42.00 ± 0.66 |
| BootstrappedNN | 5.60 ± 0.60 | 0.65 ± 0.02 | 26.23 ± 0.10 | 9.21 ± 0.44 | 73.38 ± 0.62 | 82.66 ± 0.28 | 90.60 ± 0.21 | 38.86 ± 0.08 |
| BootstrappedNN-MR | 2.15 ± 0.13 | 1.19 ± 0.15 | 31.27 ± 0.08 | 7.72 ± 0.28 | 63.26 ± 0.58 | 81.39 ± 0.12 | 99.85 ± 0.09 | 43.46 ± 0.07 |
| BootstrappedNN-SL | 3.93 ± 0.17 | 0.54 ± 0.01 | 25.53 ± 0.05 | 10.88 ± 0.61 | 70.64 ± 0.59 | 83.10 ± 0.07 | 95.92 ± 0.25 | 38.50 ± 0.06 |
| Dropout | 5.57 ± 1.02 | 2.35 ± 0.59 | 30.65 ± 0.23 | 17.55 ± 0.67 | 66.24 ± 0.74 | 84.38 ± 0.44 | 93.15 ± 0.36 | 39.82 ± 0.34 |
| Dropout-MR | 2.65 ± 0.08 | 1.30 ± 0.32 | 29.28 ± 0.12 | 10.16 ± 0.44 | 63.68 ± 0.60 | 80.66 ± 0.31 | 95.81 ± 0.21 | 36.84 ± 0.20 |
| Dropout-SL | 4.39 ± 1.02 | 1.89 ± 0.47 | 26.39 ± 0.17 | 13.18 ± 0.86 | 66.90 ± 0.80 | 81.41 ± 0.30 | 96.23 ± 0.25 | 36.96 ± 0.13 |
| GP | 11.49 ± 0.66 | 3.92 ± 0.74 | 46.25 ± 0.75 | 3.18 ± 0.08 | 74.95 ± 0.93 | 90.50 ± 0.46 | | |
| NeuralLinear | 2.22 ± 0.08 | 0.91 ± 0.01 | 29.91 ± 0.17 | 11.44 ± 0.11 | 75.43 ± 0.41 | 87.31 ± 0.27 | 95.18 ± 0.15 | 55.34 ± 0.42 |
| NeuralLinear-MR | **1.92 ± 0.10** | 1.30 ± 0.01 | 28.87 ± 0.14 | 13.47 ± 0.12 | 72.75 ± 0.50 | 86.02 ± 0.18 | 96.55 ± 0.25 | 54.01 ± 0.50 |
| NeuralLinear-SL | 2.42 ± 0.09 | **0.52 ± 0.01** | 27.60 ± 0.10 | 9.98 ± 0.56 | 71.11 ± 0.47 | 85.00 ± 0.09 | 94.99 ± 0.21 | 37.25 ± 0.06 |
| RMS | 6.68 ± 1.52 | 2.85 ± 0.73 | 27.74 ± 0.18 | 12.73 ± 0.73 | 69.93 ± 0.56 | 83.09 ± 0.24 | 91.55 ± 0.32 | 38.58 ± 0.19 |
| RMS-MR | 4.32 ± 1.06 | 2.36 ± 0.44 | 32.46 ± 0.57 | 10.72 ± 0.51 | 68.43 ± 0.72 | 87.90 ± 0.21 | 96.41 ± 0.28 | 43.64 ± 0.22 |
| RMS-SL | 3.29 ± 0.16 | 2.22 ± 0.87 | 28.25 ± 0.14 | 12.76 ± 0.63 | 71.50 ± 0.49 | 87.29 ± 0.16 | 96.63 ± 0.32 | 41.48 ± 0.19 |
| SGFS | 5.99 ± 1.02 | 3.82 ± 0.45 | 36.57 ± 0.80 | 29.00 ± 0.53 | 68.02 ± 0.63 | 88.73 ± 0.41 | 98.36 ± 0.29 | 40.55 ± 0.45 |
| SGFS-MR | 3.80 ± 0.46 | 1.44 ± 0.01 | 30.12 ± 0.05 | 12.49 ± 0.71 | 66.27 ± 0.72 | 76.00 ± 0.22 | 99.40 ± 0.33 | 37.33 ± 2.10 |
| SGFS-SL | 2.79 ± 0.10 | 2.14 ± 0.13 | 35.27 ± 0.27 | 43.95 ± 1.12 | 73.90 ± 1.51 | 85.15 ± 0.52 | 99.75 ± 0.38 | 49.37 ± 2.41 |
| ConstSGD | 26.37 ± 3.14 | 6.79 ± 1.42 | **22.47 ± 0.78** | 88.16 ± 2.80 | 70.09 ± 0.80 | 89.26 ± 0.29 | 96.84 ± 0.31 | 51.18 ± 1.74 |
| ConstSGD-MR | 3.10 ± 0.08 | 7.24 ± 1.07 | 23.39 ± 0.66 | 46.47 ± 1.85 | 71.25 ± 0.61 | 81.96 ± 0.46 | 96.02 ± 0.24 | 44.37 ± 1.39 |
| ConstSGD-SL | 41.94 ± 2.31 | 2.96 ± 0.79 | **21.61 ± 0.15** | 51.94 ± 3.78 | 70.24 ± 0.95 | 84.55 ± 0.34 | 96.08 ± 0.24 | 52.95 ± 1.29 |
| EpsGreedyRMS | 4.97 ± 1.04 | 2.13 ± 0.34 | 27.42 ± 0.13 | 12.36 ± 0.47 | 69.65 ± 0.70 | 83.33 ± 0.23 | 91.12 ± 0.23 | 38.55 ± 0.18 |
| EpsGreedyRMS-SL | 3.08 ± 0.15 | 1.09 ± 0.20 | 28.09 ± 0.09 | 7.93 ± 0.39 | 69.64 ± 0.61 | 87.65 ± 0.14 | 96.71 ± 0.29 | 40.07 ± 0.10 |
| EpsGreedyRMS-MR | 2.44 ± 0.15 | 1.71 ± 0.44 | 30.03 ± 0.20 | 8.07 ± 0.45 | 66.18 ± 0.57 | 85.20 ± 0.18 | 96.65 ± 0.25 | 40.32 ± 0.10 |
| LinDiagPost | 17.67 ± 0.18 | 51.29 ± 0.03 | 95.48 ± 0.02 | 9.59 ± 0.05 | **58.61 ± 0.49** | 99.72 ± 0.02 | 94.14 ± 0.02 | 99.43 ± 0.01 |
| LinDiagPost-MR | 8.64 ± 2.33 | 31.51 ± 0.03 | 65.03 ± 0.03 | 20.57 ± 0.13 | 60.62 ± 0.49 | 97.64 ± 0.03 | 98.33 ± 0.02 | 94.98 ± 0.02 |
| LinDiagPost-SL | 14.86 ± 2.12 | 15.60 ± 0.03 | 42.72 ± 0.03 | 2.04 ± 0.04 | 59.96 ± 0.67 | 94.45 ± 0.03 | 86.61 ± 0.02 | 90.31 ± 0.02 |
| LinDiagPrecPost | 9.48 ± 1.59 | 7.53 ± 0.02 | 34.40 ± 0.02 | 4.58 ± 0.04 | **58.58 ± 0.60** | 79.89 ± 0.03 | 87.03 ± 0.02 | 34.92 ± 0.02 |
| LinDiagPrecPost-MR | 16.21 ± 3.14 | 8.77 ± 0.07 | 35.69 ± 0.02 | 9.15 ± 0.10 | **59.08 ± 0.45** | 84.06 ± 0.03 | 89.63 ± 0.02 | 39.69 ± 0.02 |
| LinDiagPrecPost-SL | 13.17 ± 1.73 | 6.80 ± 0.02 | 33.85 ± 0.07 | **1.82 ± 0.06** | **58.83 ± 0.45** | **74.71 ± 0.05** | 85.02 ± 0.02 | 31.10 ± 0.02 |
| LinGreedy | 14.28 ± 1.99 | 11.32 ± 0.63 | 35.29 ± 0.11 | 2.18 ± 0.14 | **59.69 ± 0.60** | 83.03 ± 0.70 | **84.91 ± 0.02** | **30.73 ± 0.02** |
| LinGreedy ($\epsilon = 0.01$) | 3.38 ± 0.18 | 10.42 ± 0.39 | 34.59 ± 0.08 | 2.94 ± 0.12 | 59.95 ± 0.58 | 76.66 ± 0.25 | 85.08 ± 0.02 | 31.38 ± 0.02 |
| LinGreedy ($\epsilon = 0.05$) | 5.89 ± 0.06 | 12.75 ± 0.16 | 37.00 ± 0.03 | 6.57 ± 0.11 | 61.62 ± 0.43 | 75.75 ± 0.05 | 85.75 ± 0.02 | 34.06 ± 0.02 |
| LinPost | 6.12 ± 0.67 | 7.64 ± 0.02 | 34.40 ± 0.02 | 7.26 ± 0.05 | **59.14 ± 0.50** | 79.12 ± 0.03 | 87.17 ± 0.02 | 34.64 ± 0.02 |
| LinPost-MR | 7.93 ± 2.00 | 10.31 ± 0.03 | 38.64 ± 0.02 | 15.61 ± 0.10 | **59.17 ± 0.56** | 89.49 ± 0.04 | 92.66 ± 0.02 | 46.46 ± 0.02 |
| LinPost-SL | 14.34 ± 1.84 | 6.82 ± 0.02 | 33.61 ± 0.03 | 2.50 ± 0.03 | 60.02 ± 0.57 | 75.39 ± 0.04 | 85.43 ± 0.02 | 31.78 ± 0.02 |
| LinFullDiagPost | 86.80 ± 0.13 | 28.29 ± 0.02 | 73.82 ± 0.03 | 6.96 ± 0.06 | 63.22 ± 0.61 | 96.17 ± 0.02 | 91.60 ± 0.01 | 97.11 ± 0.01 |
| LinFullDiagPost-MR | 2.39 ± 0.08 | 14.24 ± 0.02 | 37.59 ± 0.02 | 10.25 ± 0.11 | 62.87 ± 0.42 | 85.97 ± 0.04 | 91.48 ± 0.02 | 57.15 ± 0.02 |
| LinFullDiagPost-SL | 2.24 ± 0.10 | 12.04 ± 0.04 | 37.08 ± 0.06 | 10.92 ± 0.45 | 62.56 ± 0.51 | 81.71 ± 0.09 | 90.90 ± 0.02 | 53.25 ± 0.02 |
| LinFullDiagPrecPost | 3.47 ± 0.36 | 7.34 ± 0.03 | 34.04 ± 0.02 | 4.04 ± 0.05 | 60.63 ± 0.44 | 77.00 ± 0.03 | 86.00 ± 0.02 | 32.48 ± 0.02 |
| LinFullDiagPrecPost-MR | 2.90 ± 0.34 | 7.88 ± 0.03 | 34.24 ± 0.03 | 7.74 ± 0.06 | 60.65 ± 0.50 | 77.90 ± 0.03 | 86.26 ± 0.02 | 32.91 ± 0.02 |
| LinFullDiagPrecPost-SL | 2.65 ± 0.14 | 6.84 ± 0.02 | 33.97 ± 0.06 | 4.99 ± 0.18 | 60.99 ± 0.55 | 75.81 ± 0.03 | 86.17 ± 0.02 | 31.78 ± 0.02 |
| LinFullPost | 2.37 ± 0.25 | 7.34 ± 0.02 | 34.00 ± 0.02 | 5.66 ± 0.04 | 61.87 ± 0.44 | 76.80 ± 0.03 | 86.14 ± 0.01 | 32.56 ± 0.02 |
| LinFullPost-MR | **1.82 ± 0.15** | 7.35 ± 0.02 | 34.27 ± 0.02 | 7.85 ± 0.07 | 60.76 ± 0.46 | 77.89 ± 0.03 | 86.74 ± 0.02 | 32.89 ± 0.02 |
| LinFullPost-SL | 2.62 ± 0.27 | 6.90 ± 0.02 | 33.91 ± 0.02 | 5.32 ± 0.07 | 60.89 ± 0.47 | 76.33 ± 0.03 | 86.47 ± 0.02 | 32.06 ± 0.02 |
| ParamNoise | 2.77 ± 0.15 | 1.47 ± 0.17 | 26.81 ± 0.10 | 19.04 ± 0.78 | 68.92 ± 0.53 | 87.55 ± 0.09 | 95.43 ± 0.07 | 39.20 ± 0.07 |
| ParamNoise-MR | 2.31 ± 0.11 | 1.76 ± 0.18 | 28.20 ± 0.11 | 20.25 ± 0.41 | 70.25 ± 0.64 | 86.57 ± 0.13 | 95.44 ± 0.11 | 40.46 ± 0.08 |
| ParamNoise-SL | 2.49 ± 0.09 | 1.73 ± 0.24 | 25.63 ± 0.09 | 10.62 ± 0.64 | 66.75 ± 0.54 | 81.51 ± 0.13 | 96.34 ± 0.28 | 35.75 ± 0.05 |
| Uniform | 100.00 ± 0.15 | 100.00 ± 0.03 | 100.00 ± 0.01 | 100.00 ± 1.48 | 100.00 ± 1.01 | 100.00 ± 0.02 | 100.00 ± 0.01 | 100.00 ± 0.01 |

Table 7: Simple regret incurred by models using algorithms in Section 3 on the bandits described in Section A. Simple regret was approximated by averaging the regret over the final 500 steps. Values reported are the mean over 50 independent trials with standard error of the mean. Normalized with respect to the performance of Uniform.

| | Mushroom | Statlog | Covertype | Financial | Jester | Adult | Song | Census |
|---|---|---|---|---|---|---|---|---|
| AlphaDivergence (1) | 51.57 ± 0.31 | 17.48 ± 1.51 | 30.71 ± 1.41 | 19.45 ± 0.80 | 68.37 ± 0.95 | 94.25 ± 0.19 | 97.44 ± 0.38 | 67.59 ± 0.65 |
| AlphaDivergence | 51.20 ± 0.32 | 13.32 ± 1.34 | 33.71 ± 1.72 | 26.86 ± 1.22 | 67.35 ± 0.91 | 94.36 ± 0.21 | 96.94 ± 0.39 | 63.62 ± 0.98 |
| AlphaDivergence-SL | 51.24 ± 0.36 | 22.47 ± 1.08 | 59.61 ± 0.31 | 60.74 ± 5.12 | 64.64 ± 0.86 | 93.57 ± 0.26 | 98.38 ± 0.20 | 68.01 ± 0.37 |
| BBB | **0.63 ± 0.17** | 12.47 ± 1.59 | 56.84 ± 2.21 | 19.00 ± 1.37 | 66.76 ± 0.82 | 93.57 ± 0.64 | 96.86 ± 0.42 | 64.62 ± 2.88 |
| BBB-MR | 8.94 ± 2.62 | 24.91 ± 0.28 | 60.37 ± 0.59 | 46.47 ± 3.57 | 63.18 ± 0.85 | 94.30 ± 0.43 | 95.91 ± 0.37 | 68.51 ± 1.07 |
| BBB-SL | 0.82 ± 0.16 | 1.26 ± 0.26 | 34.62 ± 0.75 | 7.46 ± 0.57 | 68.79 ± 0.93 | 87.27 ± 0.84 | 95.76 ± 0.34 | 36.74 ± 0.59 |
| BootstrappedNN | 2.43 ± 0.49 | 0.27 ± 0.04 | 22.47 ± 0.25 | 3.29 ± 0.36 | 70.29 ± 0.73 | 79.03 ± 0.40 | 87.63 ± 0.42 | 33.94 ± 0.32 |
| BootstrappedNN-MR | 1.26 ± 0.17 | 0.67 ± 0.10 | 29.81 ± 0.36 | 3.37 ± 0.32 | 59.96 ± 0.74 | 78.28 ± 0.30 | 99.14 ± 0.24 | 39.29 ± 0.37 |
| BootstrappedNN-SL | 1.44 ± 0.25 | **0.26 ± 0.03** | 21.38 ± 0.35 | 6.76 ± 0.64 | 67.17 ± 0.68 | 79.69 ± 0.31 | 95.78 ± 0.33 | 34.72 ± 0.31 |
| Dropout | 2.59 ± 1.09 | 1.34 ± 0.55 | 28.34 ± 0.42 | 11.42 ± 0.70 | 63.47 ± 0.80 | 82.18 ± 0.55 | 91.20 ± 0.43 | 36.25 ± 0.55 |
| Dropout-MR | **0.68 ± 0.14** | 0.67 ± 0.13 | 27.96 ± 0.36 | 6.04 ± 0.47 | 61.04 ± 0.74 | 77.55 ± 0.43 | 95.99 ± 0.32 | 34.34 ± 0.42 |
| Dropout-SL | 2.37 ± 1.07 | 1.38 ± 0.44 | 23.28 ± 0.40 | 8.70 ± 0.87 | 63.65 ± 0.78 | 77.91 ± 0.41 | 96.23 ± 0.34 | 34.58 ± 0.36 |
| GP | 10.66 ± 0.85 | 3.80 ± 0.74 | 45.65 ± 0.89 | **0.22 ± 0.07** | 74.38 ± 1.18 | 89.73 ± 0.59 | | |
| NeuralLinear | **0.46 ± 0.12** | 0.32 ± 0.04 | 25.30 ± 0.33 | 2.74 ± 0.08 | 70.67 ± 0.66 | 83.26 ± 0.46 | 94.46 ± 0.26 | 51.00 ± 0.62 |
| NeuralLinear-MR | 1.10 ± 0.16 | 0.56 ± 0.05 | 25.30 ± 0.36 | 3.98 ± 0.09 | 67.80 ± 0.69 | 82.03 ± 0.32 | 96.59 ± 0.31 | 48.44 ± 0.58 |
| NeuralLinear-SL | 0.84 ± 0.12 | **0.20 ± 0.03** | 23.89 ± 0.37 | 5.04 ± 0.64 | 67.80 ± 0.65 | 80.85 ± 0.26 | 94.44 ± 0.37 | 34.28 ± 0.36 |
| RMS | 5.02 ± 1.47 | 2.14 ± 0.75 | 23.71 ± 0.34 | 7.36 ± 0.69 | 65.68 ± 0.77 | 79.60 ± 0.40 | 88.02 ± 0.50 | 35.10 ± 0.39 |
| RMS-MR | 2.82 ± 1.07 | 1.72 ± 0.38 | 28.73 ± 0.47 | 5.73 ± 0.53 | 64.53 ± 0.88 | 84.59 ± 0.37 | 96.40 ± 0.31 | 39.75 ± 0.43 |
| RMS-SL | 1.22 ± 0.16 | 0.56 ± 0.21 | 23.00 ± 0.35 | 8.29 ± 0.61 | 68.15 ± 0.69 | 84.66 ± 0.29 | 96.22 ± 0.35 | 37.99 ± 0.45 |
| SGFS | 4.11 ± 0.83 | 2.41 ± 0.45 | 33.57 ± 0.86 | 16.76 ± 0.53 | 64.75 ± 0.85 | 82.80 ± 0.58 | 97.98 ± 0.38 | 35.92 ± 0.62 |
| SGFS-MR | 2.47 ± 0.47 | 1.05 ± 0.14 | 28.73 ± 0.32 | 6.74 ± 0.64 | 62.03 ± 0.74 | 74.08 ± 0.39 | 99.08 ± 0.38 | 35.24 ± 2.22 |
| SGFS-SL | 0.87 ± 0.13 | 0.87 ± 0.09 | 33.62 ± 0.42 | 19.12 ± 0.88 | 72.20 ± 1.64 | 82.64 ± 0.60 | 99.20 ± 0.42 | 46.39 ± 2.54 |
| ConstSGD | 19.57 ± 3.39 | 5.94 ± 1.46 | **15.63 ± 0.95** | 86.75 ± 3.92 | 68.48 ± 0.95 | 87.94 ± 0.46 | 96.42 ± 0.41 | 48.74 ± 2.01 |
| ConstSGD-MR | 0.85 ± 0.17 | 6.34 ± 1.10 | 17.25 ± 0.57 | 40.63 ± 2.73 | 67.64 ± 0.73 | 79.86 ± 0.50 | 96.07 ± 0.31 | 41.54 ± 1.57 |
| ConstSGD-SL | 37.01 ± 4.26 | 2.22 ± 0.78 | **16.15 ± 0.30** | 42.59 ± 4.82 | 69.47 ± 0.97 | 81.38 ± 0.41 | 95.90 ± 0.32 | 45.35 ± 1.75 |
| EpsGreedyRMS | 2.63 ± 1.09 | 0.55 ± 0.17 | 23.58 ± 0.34 | 5.83 ± 0.43 | 65.53 ± 0.89 | 79.31 ± 0.38 | 87.96 ± 0.40 | 34.46 ± 0.32 |
| EpsGreedyRMS-SL | 1.59 ± 0.21 | 0.62 ± 0.24 | 23.93 ± 0.32 | 3.32 ± 0.37 | 65.53 ± 0.76 | 84.35 ± 0.31 | 96.44 ± 0.40 | 37.01 ± 0.34 |
| EpsGreedyRMS-MR | 1.30 ± 0.19 | 1.26 ± 0.46 | 27.21 ± 0.43 | 3.73 ± 0.43 | 62.04 ± 0.66 | 81.89 ± 0.36 | 95.87 ± 0.34 | 37.07 ± 0.31 |
| LinDiagPost | 16.67 ± 0.40 | 37.05 ± 0.36 | 95.17 ± 0.22 | 2.08 ± 0.05 | **53.75 ± 0.51** | 99.78 ± 0.19 | 90.96 ± 0.29 | 98.77 ± 0.27 |
| LinDiagPost-MR | 6.85 ± 2.19 | 29.96 ± 0.30 | 61.52 ± 0.37 | 10.95 ± 0.24 | 55.44 ± 0.62 | 97.52 ± 0.19 | 97.38 ± 0.22 | 94.45 ± 0.31 |
| LinDiagPost-SL | 13.93 ± 2.06 | 13.97 ± 0.21 | 41.70 ± 0.31 | **0.16 ± 0.02** | 56.03 ± 0.79 | 94.24 ± 0.24 | 84.15 ± 0.28 | 90.62 ± 0.26 |
| LinDiagPrecPost | 8.56 ± 1.53 | 6.44 ± 0.18 | 32.97 ± 0.31 | 1.04 ± 0.04 | **54.31 ± 0.71** | 74.97 ± 0.31 | 84.83 ± 0.28 | 30.01 ± 0.35 |
| LinDiagPrecPost-MR | 15.55 ± 3.12 | 5.83 ± 0.15 | 33.40 ± 0.33 | 3.21 ± 0.08 | 54.92 ± 0.64 | 77.91 ± 0.26 | 86.27 ± 0.26 | 32.42 ± 0.27 |
| LinDiagPrecPost-SL | 12.16 ± 1.73 | 6.62 ± 0.16 | 33.17 ± 0.35 | **0.12 ± 0.04** | 54.35 ± 0.62 | **71.71 ± 0.37** | **83.14 ± 0.29** | **29.51 ± 0.31** |
| LinGreedy | 13.62 ± 2.04 | 9.35 ± 0.76 | 34.75 ± 0.33 | 0.50 ± 0.15 | 55.32 ± 0.82 | 80.52 ± 0.80 | **83.42 ± 0.29** | **29.10 ± 0.31** |
| LinGreedy ($\epsilon = 0.01$) | 1.74 ± 0.22 | 8.31 ± 0.27 | 33.73 ± 0.32 | 0.97 ± 0.10 | 55.87 ± 0.77 | 73.23 ± 0.37 | **83.14 ± 0.27** | 29.76 ± 0.33 |
| LinGreedy ($\epsilon = 0.05$) | 4.67 ± 0.35 | 11.54 ± 0.26 | 36.61 ± 0.34 | 5.12 ± 0.23 | 57.37 ± 0.60 | 74.01 ± 0.35 | 84.14 ± 0.29 | 32.85 ± 0.31 |
| LinPost | 5.31 ± 0.68 | 6.47 ± 0.15 | 33.02 ± 0.34 | 1.17 ± 0.04 | **54.91 ± 0.66** | 74.65 ± 0.29 | 84.76 ± 0.29 | **29.73 ± 0.35** |
| LinPost-MR | 6.67 ± 1.95 | 7.57 ± 0.19 | 34.25 ± 0.33 | 6.89 ± 0.11 | 55.27 ± 0.66 | 83.42 ± 0.28 | 88.66 ± 0.25 | 36.19 ± 0.34 |
| LinPost-SL | 13.69 ± 1.73 | 6.46 ± 0.17 | 32.50 ± 0.32 | 0.24 ± 0.03 | 56.15 ± 0.77 | **72.02 ± 0.32** | **83.44 ± 0.27** | 29.62 ± 0.29 |
| LinFullDiagPost | 83.94 ± 1.32 | 20.92 ± 0.28 | 73.04 ± 0.33 | 0.32 ± 0.03 | 57.29 ± 0.73 | 95.49 ± 0.23 | 88.14 ± 0.26 | 97.01 ± 0.26 |
| LinFullDiagPost-MR | 0.85 ± 0.13 | 12.24 ± 0.20 | 36.09 ± 0.37 | 2.70 ± 0.07 | 57.81 ± 0.65 | 82.39 ± 0.31 | 87.87 ± 0.31 | 54.49 ± 0.37 |
| LinFullDiagPost-SL | 0.69 ± 0.10 | 11.38 ± 0.22 | 36.42 ± 0.38 | 5.65 ± 0.37 | 57.18 ± 0.66 | 80.55 ± 0.25 | 87.66 ± 0.33 | 53.53 ± 0.34 |
| LinFullDiagPrecPost | 2.85 ± 0.37 | 6.44 ± 0.18 | 33.01 ± 0.31 | **0.18 ± 0.03** | 56.63 ± 0.69 | 73.58 ± 0.32 | 84.07 ± 0.26 | **29.59 ± 0.33** |
| LinFullDiagPrecPost-MR | 2.25 ± 0.34 | 6.21 ± 0.16 | 32.84 ± 0.32 | 0.56 ± 0.04 | 55.90 ± 0.67 | 73.16 ± 0.34 | **83.68 ± 0.27** | 30.01 ± 0.34 |
| LinFullDiagPrecPost-SL | 2.05 ± 0.16 | 6.14 ± 0.17 | 32.77 ± 0.33 | 1.15 ± 0.10 | 55.54 ± 0.77 | 72.89 ± 0.34 | **83.59 ± 0.30** | 30.22 ± 0.31 |
| LinFullPost | 1.75 ± 0.29 | 6.12 ± 0.15 | 33.16 ± 0.32 | 0.19 ± 0.03 | 57.43 ± 0.71 | 73.50 ± 0.27 | 84.01 ± 0.29 | **29.20 ± 0.37** |
| LinFullPost-MR | 1.27 ± 0.16 | 6.00 ± 0.18 | 33.01 ± 0.33 | 1.44 ± 0.04 | 55.72 ± 0.67 | 73.27 ± 0.26 | 84.70 ± 0.33 | **29.71 ± 0.33** |
| LinFullPost-SL | 2.11 ± 0.31 | 6.21 ± 0.18 | 32.99 ± 0.34 | 1.25 ± 0.06 | 55.94 ± 0.57 | 73.03 ± 0.27 | 84.81 ± 0.31 | **29.66 ± 0.29** |
| ParamNoise | 1.36 ± 0.20 | 0.66 ± 0.13 | 21.82 ± 0.30 | 15.32 ± 0.82 | 64.94 ± 0.75 | 84.55 ± 0.31 | 93.40 ± 0.28 | 35.97 ± 0.31 |
| ParamNoise-MR | 1.08 ± 0.17 | 0.83 ± 0.17 | 24.32 ± 0.37 | 15.45 ± 0.34 | 66.91 ± 0.83 | 83.52 ± 0.31 | 93.47 ± 0.31 | 36.79 ± 0.32 |
| ParamNoise-SL | **0.74 ± 0.17** | 0.62 ± 0.20 | 23.06 ± 0.35 | 4.98 ± 0.69 | 63.22 ± 0.64 | 78.58 ± 0.32 | 96.36 ± 0.35 | 32.80 ± 0.30 |
| Uniform | 100.00 ± 1.66 | 100.00 ± 0.27 | 100.00 ± 0.20 | 100.00 ± 1.58 | 100.00 ± 1.22 | 100.00 ± 0.17 | 100.00 ± 0.17 | 100.00 ± 0.20 |

Table 8: Elapsed time for algorithms in Section 3 on the bandits described in Section A. Values reported are the mean over 50 independent trials with standard error of the mean. Normalized with respect to the elapsed time required by RMS (which uses $t_s = 100$ and $t_f = 20$).

| | Mushroom | Statlog | Covertype | Financial | Jester | Adult | Song | Census |
|---|---|---|---|---|---|---|---|---|
| AlphaDivergence (1) | 560.20 ± 46.72 | 735.02 ± 55.50 | 258.25 ± 14.97 | 10289.67 ± 827.51 | 1663.50 ± 143.87 | 687.87 ± 51.09 | 228.12 ± 16.04 | 265.24 ± 48.45 |
| AlphaDivergence | 547.82 ± 36.53 | 746.48 ± 57.83 | 258.34 ± 14.62 | 10407.11 ± 858.31 | 1662.82 ± 145.74 | 668.12 ± 61.06 | 225.35 ± 13.57 | 263.95 ± 55.08 |
| AlphaDivergence-SL | 498.78 ± 120.20 | 568.54 ± 61.40 | 300.86 ± 57.50 | 5517.58 ± 1612.75 | 1033.02 ± 215.51 | 519.17 ± 98.25 | 210.92 ± 31.13 | 225.41 ± 42.02 |
| BBB | 264.64 ± 11.42 | 327.16 ± 19.62 | 146.61 ± 10.23 | 4569.10 ± 278.55 | 799.22 ± 71.61 | 314.85 ± 19.49 | 133.48 ± 9.53 | 159.10 ± 7.45 |
| BBB-MR | 227.28 ± 46.01 | 288.37 ± 55.36 | 61.35 ± 12.44 | 5537.82 ± 1035.35 | 814.01 ± 117.19 | 262.50 ± 43.01 | 52.16 ± 12.50 | 88.67 ± 25.53 |
| BBB-SL | 247.48 ± 33.11 | 315.44 ± 55.78 | 78.16 ± 12.39 | 6282.18 ± 1787.78 | 926.67 ± 269.69 | 294.29 ± 49.57 | 62.44 ± 5.58 | 100.42 ± 27.02 |
| BootstrappedNN | 1032.55 ± 61.06 | 1033.71 ± 63.65 | 1061.96 ± 85.07 | 973.91 ± 77.44 | 974.03 ± 75.80 | 1010.18 ± 55.32 | 901.19 ± 148.51 | 961.95 ± 10.50 |
| BootstrappedNN-MR | 91.13 ± 9.23 | 67.81 ± 9.11 | 79.04 ± 16.50 | 61.76 ± 9.45 | 66.32 ± 10.25 | 83.63 ± 11.26 | 68.00 ± 9.65 | 144.11 ± 45.00 |
| BootstrappedNN-SL | 109.20 ± 20.02 | 68.67 ± 7.26 | 82.75 ± 10.40 | 77.57 ± 23.63 | 75.14 ± 12.67 | 95.69 ± 14.08 | 81.20 ± 18.90 | 194.35 ± 74.09 |
| Dropout | 145.85 ± 7.07 | 149.43 ± 10.07 | 172.68 ± 12.85 | 132.70 ± 10.60 | 131.35 ± 6.79 | 138.04 ± 8.30 | 146.93 ± 6.96 | 206.57 ± 64.03 |
| Dropout-MR | 50.02 ± 4.34 | 35.26 ± 2.99 | 40.45 ± 3.15 | 37.52 ± 5.32 | 36.90 ± 5.58 | 37.85 ± 2.97 | 37.28 ± 1.73 | 69.41 ± 1.87 |
| Dropout-SL | 399.08 ± 30.09 | 467.46 ± 42.40 | 328.78 ± 26.29 | 562.34 ± 79.47 | 484.15 ± 57.88 | 410.87 ± 29.54 | 261.61 ± 23.71 | 244.49 ± 12.64 |
| GP | 4400.76 ± 388.68 | 5764.29 ± 365.29 | 1789.18 ± 267.01 | 8223.17 ± 589.89 | 7353.49 ± 487.87 | 4633.66 ± 384.08 | | |
| NeuralLinear | 137.23 ± 5.64 | 207.10 ± 14.55 | 213.15 ± 14.33 | 212.58 ± 15.98 | 197.54 ± 17.03 | 213.20 ± 14.92 | 235.60 ± 38.82 | 154.16 ± 4.39 |
| NeuralLinear-MR | 177.61 ± 12.09 | 266.55 ± 15.62 | 208.71 ± 16.18 | 277.01 ± 23.85 | 234.43 ± 18.30 | 261.31 ± 17.80 | 194.03 ± 16.88 | 182.18 ± 16.12 |
| NeuralLinear-SL | 147.82 ± 12.82 | 220.61 ± 13.56 | 170.30 ± 10.58 | 216.47 ± 13.36 | 199.22 ± 13.34 | 227.79 ± 17.89 | 156.28 ± 13.78 | 145.56 ± 18.73 |
| RMS | 100.00 ± 5.31 | 100.00 ± 6.81 | 100.00 ± 6.79 | 100.00 ± 8.33 | 100.00 ± 12.27 | 100.00 ± 8.05 | 100.00 ± 11.36 | 100.00 ± 1.49 |
| RMS-MR | 67.23 ± 5.32 | 51.63 ± 4.53 | 56.07 ± 3.40 | 56.33 ± 7.38 | 54.28 ± 6.73 | 62.00 ± 5.76 | 48.67 ± 3.12 | 77.37 ± 9.96 |
| RMS-SL | 107.91 ± 7.99 | 103.45 ± 10.36 | 90.61 ± 6.25 | 114.05 ± 18.99 | 106.35 ± 10.45 | 106.84 ± 10.06 | 78.64 ± 6.58 | 99.23 ± 8.94 |
| SGFS | 142.34 ± 5.55 | 153.14 ± 8.43 | 127.02 ± 11.76 | 193.66 ± 10.14 | 150.93 ± 8.58 | 135.86 ± 7.42 | 117.44 ± 20.54 | 143.18 ± 8.81 |
| SGFS-MR | 188.74 ± 48.01 | 178.01 ± 22.31 | 119.30 ± 16.79 | 256.76 ± 79.99 | 196.23 ± 31.54 | 169.18 ± 21.40 | 94.49 ± 14.08 | 132.36 ± 37.92 |
| SGFS-SL | 88.45 ± 19.15 | 71.34 ± 8.41 | 60.82 ± 11.43 | 99.20 ± 24.01 | 81.54 ± 13.08 | 76.97 ± 11.93 | 50.09 ± 9.19 | 84.99 ± 20.89 |
| ConstSGD | 110.95 ± 7.49 | 109.21 ± 7.49 | 107.34 ± 7.72 | 116.68 ± 12.45 | 111.59 ± 9.00 | 109.71 ± 9.09 | 103.29 ± 9.15 | 102.17 ± 2.72 |
| ConstSGD-MR | 38.16 ± 12.76 | 13.18 ± 1.90 | 24.87 ± 5.29 | 15.80 ± 4.68 | 14.76 ± 1.94 | 27.57 ± 4.61 | 25.02 ± 4.28 | 61.32 ± 17.14 |
| ConstSGD-SL | 73.90 ± 16.27 | 51.23 ± 5.90 | 52.37 ± 9.78 | 65.17 ± 32.75 | 56.33 ± 9.32 | 63.96 ± 10.24 | 45.71 ± 6.80 | 77.31 ± 17.14 |
| EpsGreedyRMS | 99.92 ± 4.05 | 100.64 ± 6.12 | 100.74 ± 7.09 | 89.64 ± 5.27 | 97.95 ± 12.09 | 98.15 ± 7.95 | 101.84 ± 11.89 | 100.09 ± 2.26 |
| EpsGreedyRMS-SL | 242.38 ± 13.74 | 267.58 ± 25.50 | 306.46 ± 9.59 | 131.72 ± 14.07 | 205.68 ± 20.47 | 237.59 ± 24.88 | 247.70 ± 20.25 | 237.48 ± 16.65 |
| EpsGreedyRMS-MR | 60.25 ± 8.41 | 52.85 ± 5.24 | 51.73 ± 5.03 | 53.57 ± 10.19 | 49.82 ± 5.50 | 54.96 ± 5.93 | 48.55 ± 4.73 | 82.01 ± 16.40 |
| LinDiagPost | 81.33 ± 3.25 | 21.36 ± 1.67 | 42.80 ± 2.73 | 11.94 ± 0.92 | 22.46 ± 2.64 | 53.66 ± 3.41 | 58.46 ± 9.35 | 316.91 ± 14.97 |
| LinDiagPost-MR | 84.63 ± 15.24 | 24.04 ± 4.11 | 45.32 ± 3.17 | 12.38 ± 2.29 | 19.43 ± 1.25 | 51.61 ± 3.29 | 46.01 ± 5.01 | 339.40 ± 13.69 |
| LinDiagPost-SL | 85.61 ± 10.53 | 27.27 ± 3.25 | 55.19 ± 5.90 | 11.31 ± 1.00 | 21.42 ± 2.62 | 53.65 ± 5.03 | 48.08 ± 3.23 | 345.99 ± 25.13 |
| LinDiagPrecPost | 85.47 ± 9.51 | 25.24 ± 1.97 | 59.29 ± 5.03 | 12.97 ± 1.35 | 22.99 ± 2.70 | 59.23 ± 4.60 | 69.98 ± 13.79 | 364.87 ± 2.87 |
| LinDiagPrecPost-MR | 93.74 ± 19.87 | 24.97 ± 2.05 | 54.88 ± 4.57 | 11.88 ± 1.42 | 20.25 ± 1.62 | 55.10 ± 3.40 | 49.61 ± 5.34 | 381.78 ± 17.04 |
| LinDiagPrecPost-SL | 86.35 ± 8.15 | 25.95 ± 3.96 | 55.29 ± 4.83 | 12.16 ± 1.80 | 21.61 ± 2.34 | 59.50 ± 4.63 | 54.34 ± 4.92 | 395.69 ± 23.62 |
| LinGreedy | 78.87 ± 11.13 | 21.43 ± 1.42 | 56.02 ± 3.93 | 3.52 ± 0.46 | 13.96 ± 1.61 | 38.65 ± 5.96 | 65.92 ± 9.82 | 156.59 ± 2.11 |
| LinGreedy ($\epsilon = 0.01$) | 73.55 ± 3.39 | 21.32 ± 1.77 | 55.23 ± 4.59 | 3.64 ± 0.42 | 13.74 ± 1.84 | 31.96 ± 3.30 | 64.04 ± 8.10 | 154.78 ± 3.15 |
| LinGreedy ($\epsilon = 0.05$) | 73.92 ± 2.98 | 20.00 ± 1.64 | 52.66 ± 3.54 | 3.51 ± 0.33 | 13.09 ± 1.76 | 31.93 ± 3.05 | 62.83 ± 10.37 | 151.76 ± 2.02 |
| LinPost | 93.53 ± 3.58 | 25.93 ± 2.65 | 64.87 ± 5.03 | 23.19 ± 1.53 | 38.20 ± 3.64 | 139.53 ± 8.83 | 82.80 ± 7.24 | 445.23 ± 26.78 |
| LinPost-MR | 96.52 ± 12.06 | 33.95 ± 6.89 | 58.76 ± 4.66 | 26.40 ± 3.62 | 33.74 ± 1.37 | 123.41 ± 6.37 | 61.13 ± 7.06 | 446.60 ± 17.31 |
| LinPost-SL | 98.21 ± 8.85 | 27.45 ± 3.39 | 61.25 ± 6.07 | 22.55 ± 2.42 | 36.00 ± 2.94 | 132.03 ± 10.26 | 66.34 ± 5.45 | 484.06 ± 65.42 |
| LinFullDiagPost | 80.47 ± 3.28 | 25.46 ± 2.57 | 48.37 ± 3.15 | 15.35 ± 1.42 | 24.18 ± 2.58 | 57.20 ± 4.86 | 58.59 ± 3.26 | 332.62 ± 8.30 |
| LinFullDiagPost-MR | 81.25 ± 6.02 | 27.68 ± 2.58 | 57.34 ± 6.74 | 14.34 ± 2.52 | 20.90 ± 1.90 | 56.78 ± 5.02 | 45.75 ± 3.97 | 366.06 ± 22.65 |
| LinFullDiagPost-SL | 77.71 ± 3.45 | 25.78 ± 2.46 | 54.59 ± 5.23 | 14.39 ± 1.51 | 23.10 ± 3.86 | 58.76 ± 5.36 | 44.71 ± 2.81 | 370.14 ± 25.12 |
| LinFullDiagPrecPost | 80.75 ± 3.03 | 26.72 ± 2.01 | 58.37 ± 4.05 | 15.18 ± 1.46 | 22.56 ± 2.16 | 61.24 ± 4.59 | 83.81 ± 17.49 | 361.93 ± 2.18 |
| LinFullDiagPrecPost-MR | 84.67 ± 9.11 | 27.35 ± 2.81 | 56.35 ± 6.31 | 14.58 ± 2.03 | 22.05 ± 1.70 | 59.77 ± 5.55 | 53.34 ± 6.68 | 386.40 ± 23.35 |
| LinFullDiagPrecPost-SL | 77.85 ± 4.66 | 25.13 ± 2.30 | 53.79 ± 4.76 | 13.68 ± 0.90 | 22.69 ± 2.44 | 57.97 ± 3.79 | 51.95 ± 4.23 | 383.85 ± 26.36 |
| LinFullPost | 94.41 ± 3.70 | 27.22 ± 1.88 | 65.02 ± 4.04 | 26.00 ± 2.05 | 40.23 ± 3.46 | 141.53 ± 9.38 | 89.72 ± 14.85 | 446.91 ± 25.61 |
| LinFullPost-MR | 99.09 ± 5.90 | 28.90 ± 2.67 | 62.19 ± 4.79 | 29.87 ± 3.17 | 38.99 ± 3.01 | 141.13 ± 6.98 | 65.31 ± 3.70 | 474.35 ± 21.76 |
| LinFullPost-SL | 99.33 ± 5.56 | 29.80 ± 2.58 | 63.71 ± 5.07 | 28.95 ± 3.94 | 38.72 ± 2.50 | 142.55 ± 6.81 | 65.01 ± 2.90 | 475.69 ± 15.56 |
| ParamNoise | 145.32 ± 7.94 | 170.52 ± 13.40 | 129.23 ± 12.58 | 228.62 ± 20.71 | 185.53 ± 22.58 | 150.22 ± 11.55 | 119.95 ± 12.29 | 111.20 ± 2.67 |
| ParamNoise-MR | 62.50 ± 13.75 | 42.40 ± 6.77 | 41.12 ± 8.23 | 60.75 ± 9.18 | 50.05 ± 6.11 | 52.95 ± 7.74 | 34.72 ± 1.44 | 75.26 ± 22.61 |
| ParamNoise-SL | 33.49 ± 2.28 | 13.23 ± 1.56 | 22.01 ± 1.51 | 16.13 ± 3.81 | 16.29 ± 2.59 | 25.40 ± 1.65 | 25.33 ± 4.22 | 57.70 ± 8.43 |
| Uniform | **0.16 ± 0.01** | **0.21 ± 0.02** | **0.22 ± 0.01** | **0.18 ± 0.03** | **0.19 ± 0.02** | **0.17 ± 0.02** | **0.18 ± 0.08** | **0.13 ± 0.01** |

Table 9: Cumulative regret incurred on the Wheel Bandit problem with increasing values of $\delta$. Values reported are the mean over 50 independent trials with standard error of the mean. Normalized with respect to the performance of Uniform.

| | $\delta = 0.5$ | $\delta = 0.7$ | $\delta = 0.9$ | $\delta = 0.95$ | $\delta = 0.99$ |
|---|---|---|---|---|---|
| AlphaDivergence (1) | 123.93 ± 0.08 | 123.65 ± 0.13 | 120.78 ± 0.59 | 115.21 ± 1.14 | 103.17 ± 0.90 |
| AlphaDivergence | 123.71 ± 0.07 | 123.71 ± 0.13 | 119.45 ± 1.15 | 117.44 ± 0.79 | 103.72 ± 0.75 |
| AlphaDivergence-SL | 11.71 ± 0.56 | 12.12 ± 0.45 | 65.16 ± 5.41 | 98.03 ± 1.58 | 101.76 ± 0.61 |
| BBB | 8.36 ± 3.59 | 19.51 ± 5.36 | 74.67 ± 6.22 | 102.35 ± 3.30 | 102.85 ± 0.74 |
| BBB-MR | 1.37 ± 0.07 | 3.32 ± 0.80 | 34.42 ± 5.50 | 59.04 ± 5.59 | 97.38 ± 2.66 |
| BBB-SL | 8.85 ± 3.68 | 8.92 ± 3.57 | 55.29 ± 6.58 | 89.31 ± 4.77 | 101.89 ± 1.05 |
| BootstrappedNN | 5.87 ± 2.65 | 28.47 ± 5.29 | 90.62 ± 5.06 | 102.59 ± 3.42 | 101.76 ± 0.96 |
| BootstrappedNN-MR | 33.00 ± 2.05 | 33.69 ± 2.65 | 45.62 ± 3.10 | 62.79 ± 3.92 | 87.27 ± 2.69 |
| BootstrappedNN-SL | 6.16 ± 2.67 | 29.00 ± 6.23 | 73.01 ± 5.46 | 97.49 ± 3.79 | 100.46 ± 1.34 |
| Dropout | 43.83 ± 4.72 | 60.62 ± 4.74 | 101.12 ± 3.77 | 108.21 ± 2.43 | 104.25 ± 0.76 |
| Dropout-MR | 7.89 ± 1.51 | 9.03 ± 2.58 | 36.58 ± 3.62 | 63.12 ± 4.26 | 98.68 ± 1.59 |
| Dropout-SL | 23.12 ± 3.54 | 51.07 ± 5.99 | 97.82 ± 4.41 | 102.17 ± 3.57 | 103.03 ± 0.75 |
| GP | 3.51 ± 0.62 | 4.82 ± 0.99 | 36.06 ± 3.53 | 65.31 ± 2.42 | 96.22 ± 1.47 |
| NeuralLinear | 1.10 ± 0.02 | 1.77 ± 0.03 | **4.32 ± 0.11** | 11.42 ± 0.97 | 52.64 ± 2.04 |
| NeuralLinear-MR | **0.95 ± 0.02** | **1.60 ± 0.03** | 4.65 ± 0.18 | **9.56 ± 0.36** | 49.63 ± 2.41 |
| NeuralLinear-SL | 1.36 ± 0.04 | 8.70 ± 3.55 | 21.93 ± 4.57 | 35.97 ± 4.08 | 76.23 ± 3.15 |
| RMS | 25.10 ± 4.73 | 44.89 ± 6.19 | 89.85 ± 4.27 | 104.72 ± 3.19 | 104.23 ± 1.07 |
| RMS-MR | 9.05 ± 1.82 | 19.42 ± 3.73 | 50.87 ± 4.49 | 74.34 ± 4.46 | 98.16 ± 1.88 |
| RMS-SL | 13.49 ± 3.94 | 26.88 ± 4.84 | 77.70 ± 6.22 | 94.84 ± 4.50 | 101.46 ± 1.20 |
| SGFS | 7.59 ± 1.10 | 10.40 ± 1.42 | 19.31 ± 1.60 | 25.55 ± 1.53 | 45.55 ± 1.49 |
| SGFS-MR | 6.04 ± 1.01 | 18.18 ± 4.29 | 42.45 ± 3.86 | 62.34 ± 3.41 | 87.12 ± 2.44 |
| SGFS-SL | 5.96 ± 0.90 | 8.88 ± 1.20 | 18.72 ± 1.73 | 33.52 ± 1.75 | 77.44 ± 2.58 |
| ConstSGD | 22.24 ± 5.11 | 33.50 ± 5.64 | 86.01 ± 5.77 | 111.36 ± 2.18 | 104.33 ± 0.78 |
| ConstSGD-MR | 25.30 ± 3.55 | 37.88 ± 3.95 | 78.74 ± 4.09 | 86.29 ± 3.45 | 97.55 ± 1.90 |
| ConstSGD-SL | 33.54 ± 5.24 | 51.14 ± 6.79 | 97.80 ± 4.80 | 109.05 ± 2.72 | 104.53 ± 0.62 |
| EpsGreedyRMS | 5.18 ± 0.85 | 27.33 ± 4.95 | 73.30 ± 4.46 | 85.43 ± 4.04 | 101.42 ± 1.45 |
| EpsGreedyRMS-SL | 2.13 ± 0.66 | 7.44 ± 2.88 | 49.63 ± 4.26 | 67.10 ± 4.02 | 95.68 ± 2.20 |
| EpsGreedyRMS-MR | 10.36 ± 2.21 | 16.96 ± 3.50 | 63.07 ± 5.45 | 80.88 ± 4.79 | 97.69 ± 2.28 |
| LinDiagPost | 1.12 ± 0.03 | 1.80 ± 0.08 | 5.06 ± 0.14 | **8.99 ± 0.33** | **37.77 ± 2.18** |
| LinDiagPost-MR | 9.53 ± 0.45 | 6.33 ± 0.19 | 11.81 ± 0.83 | 20.81 ± 1.88 | 51.19 ± 2.09 |
| LinDiagPost-SL | 44.19 ± 4.41 | 40.65 ± 4.24 | 54.32 ± 3.34 | 66.09 ± 3.52 | 81.29 ± 2.36 |
| LinDiagPrecPost | 1.20 ± 0.04 | 1.85 ± 0.07 | 5.43 ± 0.19 | 9.84 ± 0.33 | **40.96 ± 2.25** |
| LinDiagPrecPost-MR | 14.87 ± 2.81 | 21.68 ± 4.03 | 57.41 ± 5.66 | 72.57 ± 4.08 | 87.98 ± 2.73 |
| LinDiagPrecPost-SL | 41.50 ± 4.31 | 34.41 ± 3.68 | 61.86 ± 4.33 | 79.32 ± 4.45 | 93.23 ± 2.39 |
| LinGreedy | 65.89 ± 4.90 | 71.71 ± 4.31 | 108.86 ± 3.10 | 102.80 ± 3.06 | 104.80 ± 0.91 |
| LinGreedy ($\epsilon = 0.01$) | 11.67 ± 1.03 | 17.28 ± 1.29 | 46.14 ± 2.37 | 70.01 ± 3.52 | 96.54 ± 1.57 |
| LinGreedy ($\epsilon = 0.05$) | 7.86 ± 0.27 | 9.58 ± 0.35 | 19.42 ± 0.78 | 33.06 ± 2.06 | 74.17 ± 1.63 |
| LinPost | 1.10 ± 0.04 | 1.73 ± 0.06 | 4.95 ± 0.17 | 10.31 ± 0.49 | 42.59 ± 2.05 |
| LinPost-MR | 6.18 ± 0.27 | 4.99 ± 0.38 | 9.28 ± 0.61 | 13.74 ± 0.64 | 43.18 ± 1.52 |
| LinPost-SL | 41.60 ± 3.93 | 46.98 ± 4.79 | 69.06 ± 3.70 | 81.39 ± 3.26 | 93.29 ± 1.98 |
| LinFullDiagPost | 2.45 ± 0.04 | 2.83 ± 0.05 | 6.18 ± 0.12 | 11.45 ± 0.39 | 53.63 ± 2.67 |
| LinFullDiagPost-MR | 9.54 ± 0.72 | 11.13 ± 1.56 | 43.59 ± 5.03 | 50.67 ± 4.79 | 91.84 ± 2.60 |
| LinFullDiagPost-SL | 44.95 ± 4.66 | 58.60 ± 5.40 | 86.66 ± 4.97 | 84.89 ± 4.63 | 91.02 ± 2.50 |
| LinFullDiagPrecPost | 1.55 ± 0.03 | 2.06 ± 0.06 | 5.38 ± 0.14 | 9.83 ± 0.39 | 57.42 ± 2.47 |
| LinFullDiagPrecPost-MR | 1.98 ± 0.05 | 2.33 ± 0.06 | 5.45 ± 0.11 | **9.36 ± 0.23** | 42.74 ± 2.18 |
| LinFullDiagPrecPost-SL | 27.93 ± 3.74 | 45.72 ± 4.66 | 69.33 ± 4.50 | 91.37 ± 3.97 | 95.43 ± 2.16 |
| LinFullPost | 1.63 ± 0.03 | 2.11 ± 0.05 | 5.28 ± 0.16 | 10.65 ± 0.57 | 57.86 ± 2.84 |
| LinFullPost-MR | 8.42 ± 1.26 | 11.69 ± 2.16 | 31.00 ± 2.61 | 52.58 ± 3.50 | 83.83 ± 2.35 |
| LinFullPost-SL | 41.68 ± 5.02 | 38.21 ± 4.18 | 73.95 ± 5.74 | 81.44 ± 5.28 | 90.81 ± 2.64 |
| ParamNoise | 22.39 ± 3.27 | 48.22 ± 4.75 | 94.57 ± 4.34 | 110.10 ± 2.47 | 103.16 ± 0.89 |
| ParamNoise-MR | 1.54 ± 0.20 | 3.30 ± 0.65 | 18.02 ± 2.27 | 38.93 ± 3.20 | 88.25 ± 2.07 |
| ParamNoise-SL | 12.07 ± 1.76 | 18.67 ± 2.40 | 58.87 ± 4.49 | 61.55 ± 2.92 | 95.13 ± 2.21 |
| Uniform | 100.00 ± 0.08 | 100.00 ± 0.09 | 100.00 ± 0.25 | 100.00 ± 0.37 | 100.00 ± 0.78 |

Table 10: Simple regret incurred on the Wheel Bandit problem with increasing values of $\delta$. Simple regret was approximated by averaging the regret over the final 500 steps. Values reported are the mean over 50 independent trials with standard error of the mean. Normalized with respect to the performance of Uniform.

| | $\delta = 0.5$ | $\delta = 0.7$ | $\delta = 0.9$ | $\delta = 0.95$ | $\delta = 0.99$ |
|---|---|---|---|---|---|
| AlphaDivergence (1) | $122.82 \pm 0.36$ | $124.88 \pm 0.74$ | $121.83 \pm 2.03$ | $118.13 \pm 3.03$ | $114.60 \pm 4.70$ |
| AlphaDivergence | $123.78 \pm 0.44$ | $126.05 \pm 0.96$ | $117.76 \pm 2.15$ | $121.01 \pm 2.66$ | $110.49 \pm 5.07$ |
| AlphaDivergence-SL | $1.79 \pm 0.15$ | $2.80 \pm 0.36$ | $56.95 \pm 6.18$ | $93.41 \pm 3.37$ | $105.30 \pm 4.55$ |
| BBB | $6.31 \pm 3.58$ | $15.35 \pm 5.63$ | $68.68 \pm 7.40$ | $105.63 \pm 4.34$ | $101.63 \pm 4.54$ |
| BBB-MR | $0.60 \pm 0.09$ | $1.45 \pm 0.61$ | $27.03 \pm 6.19$ | $56.64 \pm 6.36$ | $102.96 \pm 5.98$ |
| BBB-SL | $6.48 \pm 3.63$ | $6.17 \pm 3.61$ | $47.39 \pm 7.51$ | $88.12 \pm 6.22$ | $110.58 \pm 5.25$ |
| BootstrappedNN | $4.25 \pm 2.53$ | $24.19 \pm 5.41$ | $88.95 \pm 5.88$ | $104.31 \pm 4.23$ | $105.10 \pm 4.11$ |
| BootstrappedNN-MR | $25.92 \pm 2.33$ | $28.73 \pm 2.90$ | $38.76 \pm 3.65$ | $57.72 \pm 5.11$ | $89.08 \pm 5.56$ |
| BootstrappedNN-SL | $4.83 \pm 2.73$ | $24.65 \pm 6.52$ | $67.63 \pm 6.09$ | $93.62 \pm 5.61$ | $103.26 \pm 5.30$ |
| Dropout | $43.07 \pm 4.77$ | $60.47 \pm 4.82$ | $102.49 \pm 3.89$ | $113.84 \pm 3.74$ | $107.98 \pm 4.89$ |
| Dropout-MR | $6.57 \pm 1.48$ | $6.37 \pm 2.53$ | $35.02 \pm 3.94$ | $59.45 \pm 4.74$ | $102.12 \pm 4.76$ |
| Dropout-SL | $19.96 \pm 3.63$ | $49.42 \pm 6.35$ | $99.31 \pm 5.50$ | $104.06 \pm 4.66$ | $103.39 \pm 5.04$ |
| GP | $2.80 \pm 0.57$ | $3.86 \pm 1.05$ | $34.61 \pm 3.80$ | $66.53 \pm 3.21$ | $96.94 \pm 4.65$ |
| NeuralLinear | $\mathbf{0.31 \pm 0.03}$ | $\mathbf{0.68 \pm 0.07}$ | $\mathbf{2.18 \pm 0.13}$ | $5.44 \pm 0.73$ | $46.42 \pm 3.45$ |
| NeuralLinear-MR | $\mathbf{0.33 \pm 0.04}$ | $\mathbf{0.79 \pm 0.07}$ | $\mathbf{2.17 \pm 0.14}$ | $\mathbf{4.08 \pm 0.20}$ | $35.89 \pm 2.98$ |
| NeuralLinear-SL | $0.39 \pm 0.04$ | $6.65 \pm 3.47$ | $16.63 \pm 4.73$ | $28.73 \pm 4.85$ | $71.53 \pm 5.90$ |
| RMS | $22.98 \pm 4.67$ | $43.58 \pm 6.51$ | $88.70 \pm 4.88$ | $109.31 \pm 4.48$ | $105.06 \pm 4.31$ |
| RMS-MR | $7.75 \pm 1.86$ | $16.53 \pm 3.48$ | $47.26 \pm 4.76$ | $72.09 \pm 5.40$ | $109.20 \pm 4.62$ |
| RMS-SL | $11.01 \pm 3.97$ | $22.29 \pm 4.70$ | $74.99 \pm 7.16$ | $92.27 \pm 5.07$ | $103.35 \pm 4.79$ |
| SGFS | $4.57 \pm 0.96$ | $7.15 \pm 1.39$ | $13.34 \pm 1.73$ | $17.76 \pm 1.65$ | $35.03 \pm 2.68$ |
| SGFS-MR | $1.96 \pm 0.35$ | $14.70 \pm 4.24$ | $39.18 \pm 4.39$ | $59.62 \pm 4.17$ | $87.27 \pm 4.54$ |
| SGFS-SL | $3.09 \pm 0.54$ | $4.99 \pm 1.05$ | $10.79 \pm 1.46$ | $24.07 \pm 2.26$ | $73.50 \pm 4.91$ |
| ConstSGD | $20.40 \pm 5.16$ | $30.44 \pm 5.93$ | $87.26 \pm 6.37$ | $114.11 \pm 3.22$ | $109.77 \pm 4.61$ |
| ConstSGD-MR | $21.85 \pm 3.66$ | $36.00 \pm 4.11$ | $75.03 \pm 4.74$ | $86.91 \pm 4.31$ | $94.72 \pm 4.78$ |
| ConstSGD-SL | $32.38 \pm 5.33$ | $49.48 \pm 7.05$ | $97.36 \pm 4.87$ | $109.47 \pm 3.49$ | $108.68 \pm 5.24$ |
| EpsGreedyRMS | $1.59 \pm 0.67$ | $20.47 \pm 5.20$ | $64.39 \pm 5.52$ | $81.85 \pm 5.48$ | $104.21 \pm 5.09$ |
| EpsGreedyRMS-SL | $0.96 \pm 0.54$ | $5.46 \pm 2.83$ | $46.09 \pm 4.73$ | $64.99 \pm 4.30$ | $102.72 \pm 4.31$ |
| EpsGreedyRMS-MR | $7.96 \pm 2.01$ | $13.70 \pm 3.65$ | $60.71 \pm 5.83$ | $83.15 \pm 6.07$ | $96.78 \pm 4.99$ |
| LinDiagPost | $0.51 \pm 0.05$ | $1.06 \pm 0.10$ | $3.35 \pm 0.30$ | $5.41 \pm 0.50$ | $\mathbf{26.33 \pm 3.28}$ |
| LinDiagPost-MR | $0.99 \pm 0.07$ | $0.86 \pm 0.07$ | $3.11 \pm 0.21$ | $10.13 \pm 1.91$ | $31.16 \pm 2.34$ |
| LinDiagPost-SL | $40.88 \pm 4.54$ | $37.69 \pm 4.39$ | $48.01 \pm 3.85$ | $62.42 \pm 4.36$ | $71.54 \pm 4.22$ |
| LinDiagPrecPost | $0.41 \pm 0.04$ | $0.87 \pm 0.10$ | $3.57 \pm 0.30$ | $5.17 \pm 0.37$ | $\mathbf{29.49 \pm 3.02}$ |
| LinDiagPrecPost-MR | $10.72 \pm 3.16$ | $18.06 \pm 4.14$ | $59.68 \pm 6.45$ | $72.61 \pm 4.79$ | $92.59 \pm 6.08$ |
| LinDiagPrecPost-SL | $39.63 \pm 4.34$ | $29.90 \pm 3.88$ | $59.13 \pm 5.05$ | $79.67 \pm 5.62$ | $89.95 \pm 5.20$ |
| LinGreedy | $66.59 \pm 5.02$ | $73.06 \pm 4.55$ | $108.56 \pm 3.65$ | $105.01 \pm 3.59$ | $105.19 \pm 4.14$ |
| LinGreedy ($\epsilon = 0.01$) | $2.36 \pm 0.19$ | $2.55 \pm 0.19$ | $16.11 \pm 2.32$ | $45.86 \pm 4.79$ | $88.92 \pm 4.72$ |
| LinGreedy ($\epsilon = 0.05$) | $5.53 \pm 0.19$ | $6.07 \pm 0.24$ | $8.49 \pm 0.47$ | $12.65 \pm 1.12$ | $57.62 \pm 3.57$ |
| LinPost | $0.42 \pm 0.05$ | $0.97 \pm 0.10$ | $2.93 \pm 0.26$ | $5.54 \pm 0.44$ | $32.01 \pm 3.06$ |
| LinPost-MR | $0.70 \pm 0.06$ | $0.99 \pm 0.10$ | $3.08 \pm 0.22$ | $4.85 \pm 0.27$ | $\mathbf{25.42 \pm 1.81}$ |
| LinPost-SL | $35.81 \pm 3.86$ | $37.32 \pm 4.76$ | $59.50 \pm 4.75$ | $69.56 \pm 4.47$ | $94.85 \pm 5.05$ |
| LinFullDiagPost | $1.13 \pm 0.06$ | $1.58 \pm 0.09$ | $3.16 \pm 0.22$ | $5.62 \pm 0.41$ | $39.24 \pm 3.33$ |
| LinFullDiagPost-MR | $2.07 \pm 0.64$ | $4.76 \pm 1.66$ | $40.16 \pm 5.60$ | $44.12 \pm 5.35$ | $93.98 \pm 5.62$ |
| LinFullDiagPost-SL | $42.92 \pm 5.04$ | $58.51 \pm 5.82$ | $89.72 \pm 5.48$ | $86.56 \pm 5.33$ | $86.16 \pm 4.75$ |
| LinFullDiagPrecPost | $0.72 \pm 0.07$ | $0.91 \pm 0.09$ | $3.43 \pm 0.25$ | $\mathbf{4.65 \pm 0.43}$ | $45.19 \pm 3.59$ |
| LinFullDiagPrecPost-MR | $0.65 \pm 0.06$ | $0.98 \pm 0.11$ | $2.94 \pm 0.24$ | $4.59 \pm 0.30$ | $\mathbf{26.57 \pm 2.51}$ |
| LinFullDiagPrecPost-SL | $25.86 \pm 3.97$ | $44.24 \pm 4.63$ | $67.12 \pm 4.90$ | $94.04 \pm 4.75$ | $94.17 \pm 4.72$ |
| LinFullPost | $0.65 \pm 0.06$ | $1.09 \pm 0.09$ | $3.53 \pm 0.22$ | $\mathbf{4.46 \pm 0.33}$ | $45.04 \pm 4.01$ |
| LinFullPost-MR | $3.47 \pm 1.20$ | $7.45 \pm 2.30$ | $20.32 \pm 2.65$ | $41.85 \pm 3.82$ | $79.54 \pm 4.56$ |
| LinFullPost-SL | $39.42 \pm 5.33$ | $37.20 \pm 4.40$ | $75.61 \pm 6.46$ | $83.50 \pm 6.18$ | $96.19 \pm 5.73$ |
| ParamNoise | $18.75 \pm 3.10$ | $47.23 \pm 4.97$ | $92.26 \pm 4.95$ | $110.49 \pm 3.72$ | $110.64 \pm 5.21$ |
| ParamNoise-MR | $0.38 \pm 0.04$ | $1.36 \pm 0.56$ | $10.46 \pm 2.00$ | $28.77 \pm 3.72$ | $89.13 \pm 4.79$ |
| ParamNoise-SL | $7.54 \pm 1.80$ | $11.45 \pm 2.22$ | $52.55 \pm 4.95$ | $59.05 \pm 3.66$ | $97.85 \pm 4.86$ |
| Uniform | $100.00 \pm 0.45$ | $100.00 \pm 0.78$ | $100.00 \pm 1.18$ | $100.00 \pm 2.21$ | $100.00 \pm 4.21$ |

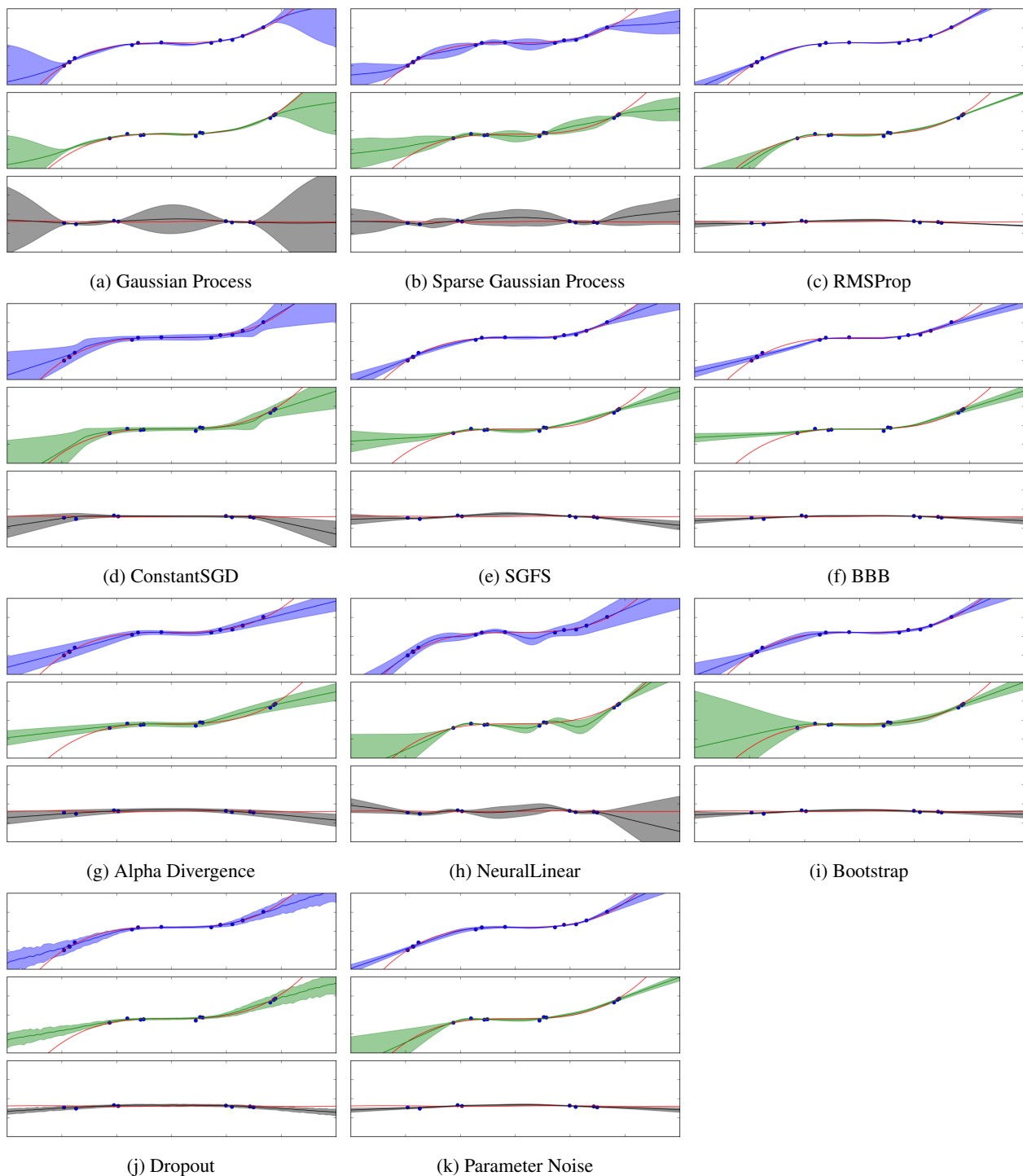

Figure 6: We qualitatively compare plots of the sample distribution from various methods, similarly to Hernández-Lobato et al. (2016). We plot the mean and standard deviation of 100 samples drawn from each method conditioned on a small set of observations with three outputs (two are from the same underlying function and thus strongly correlated while the third (bottom) is independent). The true underlying functions are plotted in red.

# A   REAL-WORLD DATASETS

**Mushroom.** The Mushroom Dataset (Schlimmer, 1981) contains 22 attributes per mushroom, and two classes: poisonous and safe. As in Blundell et al. (2015), we create a bandit problem where the agent must decide whether to eat or not a given mushroom. Eating a safe mushroom provides reward +5. Eating a poisonous mushroom delivers reward +5 with probability 1/2 and reward -35 otherwise. If the agent does not eat a mushroom, then the reward is 0. We set $n = 50000$.

**Statlog.** The Shuttle Statlog Dataset (Asuncion & Newman, 2007) provides the value of $d = 9$ indicators during a space shuttle flight, and the goal is to predict the state of the radiator subsystem of the shuttle. There are $k = 7$ possible states, and if the agent selects the right state, then reward 1 is generated. Otherwise, the agent obtains no reward ($r = 0$). The most interesting aspect of the dataset is that one action is the optimal one in 80% of the cases, and some algorithms may commit to this action instead of further exploring. In this case, $n = 43500$.

**Covertype.** The Covertype Dataset (Asuncion & Newman, 2007) classifies the cover type of northern Colorado forest areas in $k = 7$ classes, based on $d = 54$ features, including elevation, slope, aspect, and soil type. Again, the agent obtains reward 1 if the correct class is selected, and 0 otherwise. We run the bandit for $n = 150000$.

**Financial.** We created the Financial Dataset by pulling the stock prices of $d = 21$ publicly traded companies in NYSE and Nasdaq, for the last 14 years ($n = 3713$). For each day, the context was the price difference between the beginning and end of the session for each stock. We synthetically created the arms, to be a linear combination of the contexts, representing $k = 8$ different potential portfolios. By far, this was the smallest dataset, and many algorithms over-explored at the beginning with no time to amortize their investment (Thompson Sampling does not account for the horizon).

**Jester.** We create a recommendation system bandit problem as follows. The Jester Dataset (Goldberg et al., 2001) provides continuous ratings in $[-10, 10]$ for 100 jokes from 73421 users. We find a complete subset of $n = 19181$ users rating all 40 jokes. Following Riquelme et al. (2017), we take $d = 32$ of the ratings as the context of the user, and $k = 8$ as the arms. The agent recommends one joke, and obtains the reward corresponding to the rating of the user for the selected joke.

**Adult.** The Adult Dataset (Kohavi, 1996; Asuncion & Newman, 2007) comprises personal information from the US Census Bureau database, and the standard prediction task is to determine if a person makes over $50K a year or not. However, we consider the $k = 14$ different occupations as feasible actions, based on $d = 94$ covariates (many of them binarized). As in previous datasets, the agent obtains reward 1 for making the right prediction, and 0 otherwise. We set $n = 45222$.

**Census.** The US Census (1990) Dataset (Asuncion & Newman, 2007) contains a number of personal features (age, native language, education...) which we summarize in $d = 389$ covariates, including binary dummy variables for categorical features. Our goal again is to predict the occupation of the individual among $k = 9$ classes. The agent obtains reward 1 for making the right prediction, and 0 otherwise, for each of the $n = 250000$ randomly selected data points.

**Song.** The YearPredictionMSD Dataset is a subset of the Million Song Dataset (Bertin-Mahieux et al., 2011). The goal is to predict the year a given song was released (1922-2011) based on $d = 90$ technical audio features. We divided the years in $k = 10$ contiguous year buckets containing the same number of songs, and provided decreasing Gaussian rewards as a function of the distance between the interval chosen by the agent and the one containing the year the song was actually released. We initially selected $n = 250000$ songs at random from the training set.

The Statlog, Covertype, Adult, and Census datasets were tested in Elmachtoub et al. (2017).

