# OpenReview forum: "Deep Bayesian Bandits Showdown:  An Empirical Comparison of Bayesian Deep Networks for Thompson Sampling"
_ICLR.cc/2018/Conference — Accept (Poster)_

### Official Review · AnonReviewer1 · 2017-11-26
**Benchmark most useful if accompanying code is well engineered**

**Rating:** 7
**Confidence:** 4

**Review:**

If two major questions below are answered affirmatively, I believe this article could be very good contribution to the field and deserve publication in ICLR.

In this article the authors provide a service to the community by comparing the current most used algorithms for Thompson Sampling-based contextual (parametric) bandits on clear empirical benchmark. They reimplement the key algorithms, investing time to make up for the lack of published source code for some.

After a clear exposure of the reasons why Thompson Sampling is attractive, they overview concisely the key ideas behind 7 different families of algorithms, with proper literature review. They highlight some of the subtleties of benchmarking bandit problems (or any active learning algorithms for that matter): the lack of counterfactual and hence the difference in observed datasets. They explain their benchmark framework and datasets, then briefly summarise the results for each class of algorithms. Most of the actual measures from the benchmark are provided in a lengthy appendix 12 pages appendix choke-full of graphs and tables.

It is refreshing to see an article that does not boast to offer the new "bestest-ever" algorithm in town, overcrowding a landscape, but instead tries to prune the tree of possibilities and wading through other people's inflated claims. To the authors: thank you! It is too easy to dismiss these articles as "pedestrian non-innovative groundwork": if there were more like it, our field would certainly be more readable and less novelty-prone.

Of course, there is no perfect benchmark, and like every benchmark, the choices made by the authors could be debated to no end. At least, the authors try to explain them, and the tradeoffs they faced, as clearly as possible (except for two points mentioned below), which again is too rare in our field.

Major clarifications needed:

My two key questions are:
* Is the code of good quality, with exact  reproducibility and good potential extension in a standard language (e.g. Python)? This benchmark only gets its full interest if the code is publicised and well engineered. The open-sourcing is planned, according to footnote 1, is planned -- but this should be made clearer in the main text. There is no discussion of the engineering quality, not even of the language used, and this is quite important if the authors want the community to build upon this work. The code was not submitted for review, and as such its accessibility to new contributors is unknown to this reviewer. That could be a make or break feature of this work.
* Is the hyper parameter tuning reproducible? Hyperparameter tuning should be discussed much more clearly (in the Appendix): while I appreciate the discussion page 8 of how they were frozen across datasets, "they were chosen through careful tuning" is way too short. What kind of tuning? Was it  manual, and hence not reproducible? Or was it a clear, reproducible grid search or optimiser? I thoroughly hope for the later, otherwise an unreproducible benchmark would be very

If the answers to the two questions above is "YES", then brilliant article, I am ready to increase my score. However, if either is a "NO", I am afraid that would limit to how much this benchmark will serve as a reference (as opposed to "just one interesting datapoint").


Minor improvements:
* Please proofread some obvious typos:
  - page 4 "suggesed" -> "suggested",
  - page 8 runaway math environment wreaking the end of the sentence.
  - reference "Meire Fortunato (2017)" should be  "Fortunato et al. (2017)", throughout.
* Improve readability of figures' legends, e.g. Figure 2.(b) key is un-readable.
* A simple table mapping the name of the algorithm to the corresponding article is missing. Not everyone knows what BBB and BBBN stands for.
* A measure of wall time would be needed: while computational cost is often mentioned (especially as a drawback to getting proper performance out of variational inference), it is nowhere plotted. Of course that would partly depend on the quality of the implementation, but this is somewhat mitigated if all the algorithms have been reimplemented by the authors (is that the case? please clarify).

---

> ### Author Response · Authors · 2017-12-20
> **Re: Benchmark most useful if accompanying code is well engineered**
>
> We thank the reviewer for carefully reading the manuscript and for their thoughtful feedback.
>
> To address the primary concerns:
>
> 1 - The code is written in Python and Tensorflow, and will be committed to a well-known Anonymized open source library. Currently, the code is going through third party code review within our organization and is subject to a high quality standard. We designed the implementation so that adding new algorithms and rerunning the benchmark is straightforward for an external contributor.
>
> 2 - We agree that making the hyperparameter selection reproducible is essential. To this end, we will re-run the experiments doing the following: 1) we will choose two representative datasets and apply Bayesian optimization to find parameters for each algorithm based on the results from the training datasets. Then, we will freeze these parameters for the remaining datasets and report numbers (and parameters) on these heldout datasets. We will update this post when we have revised the manuscript with the new numbers.
>
> Finally, we have fixed the typos and improved the figures' legends. We added a table mapping algorithm names to their meaning and parameters. We agree that a table showing wall clock time for each algorithm is highly informative, and we plan to add that to the revised manuscript.
>
> We confirm that the authors reimplemented all of the algorithms.

---

### Official Review · AnonReviewer2 · 2017-11-27
**An interesting comparative study of deep neural bandits (but with rather limited results analysis)**

**Rating:** 6
**Confidence:** 4

**Review:**

The paper "DEEP BAYESIAN BANDITS SHOWDOWN" proposes a comparative study about bandit approaches using deep neural networks.

While I find that such a study is a good idea, and that I was really interested by the listing of the different possibilities in the algorithms section, I regret that the experimental results given and their analysis do not allow the reader to well understand the advantages and issues of the approaches. The given discussion is not enough connected to the presented results from my point of view and it is difficult to figure out what is the basis of some conclusion.

Also, the considered algorithms are not enough described to allow the reader to have enough insights to fully understand the proposed arguments. Maybe authors should have focused on less algorithms but with more implementation details. Also, what does not help is that it is very hard to conect the names in the result table with the corresponding approaches (some abbreviations are not defined at all - BBBN or RMS for instances).

At last, the experimental protocol should be better described. For instance it is not clear on how the regret is computed : is it based on the best expectation (as done in most os classical studies) or on the best actual score of actions? The wheel bandit protocol is also rather hard to follow (and where is the results analysis?).

Other remarks:
   - It is a pitty that expectation propagation approaches have been left aside since they correspond to an important counterpart to variational ones. It would have been nice to get a comparaison of both;
   - Variational inference decsription in section algorithms is not enough developped w.r.t. the importance of this family of approaches
   - Neural Linear is strange to me. Uncertainty does not consider the neural representation of inputs ? How does it work then ?
   - That is strange that \Lambda_0 and \mu_0 do not belong to the stated asumptions in the linear methods part (ok they correspond to some  prior but it should be clearly stated)
   - Figure 1 is referenced very late (after figure 2)

---

> ### Author Response · Authors · 2017-12-20
> **RE: An interesting comparative study of deep neural bandits (but with rather limited results analysis)**
>
> First, we would like to thank the reviewer for their feedback.
>
> We acknowledge that the submitted version of the paper does not clearly connect the numerical results and our conclusions and claims. For the revision, we are focused on improving clarity. We plan to expand the discussion of the results and to add tables that summarize the relative ranking among algorithms across datasets to make comparison simpler.
>
> Moreover, we plan to extend the sections corresponding to algorithm descriptions and experimental setup. We also now include a table that explains the abbreviated algorithm names and hyperparameter settings (e.g., difference between RMS2 and RMS3, etc.).
>
> Regret is computed based on the best expected reward (as is standard). For some real datasets, the rewards were deterministic, in which case, both definitions of regret agree. We reshuffle the order of the contexts, and rerun the experiment a number of times to obtain the cumulative regret distribution and report its statistics. We now clarify this procedure in the experimental setup section.
>
> We agree that the wheel bandit protocol was not clearly explained, and we have expanded the description.
>
> We agree that expectation propagation methods are relevant to this study, so we have implemented the black-box alpha-divergence algorithm [1] and will add it to the study.
>
> NeuralLinear is based on a standard deep neural network. However, decisions are made according to a Bayesian linear regression applied to the features at the last layer of the network. Note that the last hidden layer representation determines the final output of the network via a linear function, so we can expect a representation that explains the expected value of an action with a linear model. For all the training contexts, their deep representation is computed, and then uncertainty estimates on linear parameters for each action are derived via standard formulas. Thompson sampling will sample from this distribution, say \beta_t,i at time t for action i, and the next context will be pushed through the network until the last layer, leading to its representation c_t. Then, the sampled beta’s will predict an expected value, and the action with the highest prediction will be taken. Importantly, the algorithm does not use any uncertainty estimates on the representation itself (as opposed to variational methods, for example). On the other hand, the way the algorithm handles uncertainty conditional on the representation and the linear assumption is exact, which seems to be key to its success.
>
> We will add a comment explaining the assumed prior for linear methods.
>
> [1] Hernández-Lobato, J. M., Li, Y., Rowland, M., Hernández-Lobato, D., Bui, T., and Turner, R. E. (2016). Black-box α-divergence minimization. In International Conference on Machine Learning.

---

### Official Review · AnonReviewer3 · 2017-11-29
**A large-scale comparison on some posterior estimation methods for Thompson sampling without much insight**

**Rating:** 5
**Confidence:** 5

**Review:**

This paper presents the comparison of a list of algorithms for contextual bandit with Thompson sampling subroutine. The authors compared different methods for posterior estimation for Thompson sampling. Experimental comparisons on contextual bandit settings have been performed on a simple simulation and quite a few real datasets.

The main paper + appendix are clearly written and easy to understand. The main paper itself is very incomplete. The experimental results should be summarized and presented in the main context. There is a lack of novelty of this study. Simple comparisons of different posterior estimating methods do not provide insights or guidelines for contextual bandit problem.

What's the new information provided by running such methods on different datasets? What are the newly observed advantages and disadvantages of them? What could be the fundamental reasons for the variety of behaviors on different datasets? No significant conclusions are made in this work.

Experimental results are not very convincing. There are lots of plots show linear cumulative regrets within the whole time horizon. Linear regrets represent either trivial methods or not long enough time horizon.

---

> ### Author Response · Authors · 2017-12-21
> **RE: A large-scale comparison on some posterior estimation methods for Thompson sampling without much insight**
>
> We thank the reviewer for their feedback. The reviewer raises several important concerns, which we address below.
>
> Overall, the main concerns were a lack of insightful conclusions/practical guidelines and that the paper relies too heavily on the appendix. Unfortunately, due to poor organization and writing, the insights we gained from the empirical benchmark were not made clear. We plan to significantly revise the paper for clarity. We briefly summarize our contributions and the insights we derived from the empirical results:
>
> Several recent papers claim to innovate on exploration with deep neural networks (e.g., two concurrent ICLR submissions: https://openreview.net/forum?id=ByBAl2eAZ, https://openreview.net/forum?id=rywHCPkAW). We argue that such innovations should be benchmarked against existing literature and baselines on simple decision making tasks (if the methods don’t improve on contextual bandits, how could they hope to improve in RL?). Our major contribution is this empirical comparison - a series of reproducible benchmarks with baseline implementations (all of which will be open sourced). We hope that the reviewer agrees that this empirical benchmark is a scientifically useful contribution.
>
> From the empirical benchmark, we find that:
>
> 1) Variational approaches to estimate uncertainty in neural networks are an active area of research, however, to the best of our knowledge, there is no study that systematically benchmarks variational approaches in decision-making scenarios against other state-of-the-art approaches.
>
> From our evaluation, surprisingly, we find that Bayes by Backprop (BBB) underperforms even with a linear model. We demonstrate that because the method is simultaneously learning the representation and the uncertainty level, when faced with a limited optimization budget (for online learning), slow convergence becomes a serious concern. In particular, when the fitted model is linear, we evaluate the performance of a mean field model which we we can solve in closed form for the variational objective. We find that as we increase number of training iterations for BBB, it slowly converges to the performance of this exact method (Fig 25). We also see that the difference can be much larger than the degradation due to using a mean field approximation. We plan to move this experiment to the main text and expand upon the details.
>
> This is not a problem in the supervised learning setting, where we can train until convergence. Unfortunately, in the online learning setting, this is problematic, as we cannot train for an unreasonable number of iterations at each step, so poor uncertainty estimates lead to bad decisions. Additionally, tricks to speed up convergence of BBB, such as initializing the variance parameters to a small value, distort uncertainty estimates and thus are not applicable in the online decision making setting.
>
> We believe that these insights into the problems with variational approaches are of value to the community, and highlight the need for new ways to estimate uncertainty for online scenarios (i.e., without requiring great computational power).
>
> 2) We study an algorithm, which we call NeuralLinear, that is remarkably simple, and combines two classic ideas (NNs and Bayesian linear regression). A very similar algorithm was used before in Bayesian optimization [1] and an independent ICLR submission (https://openreview.net/forum?id=Bk6qQGWRb) proposes nearly the same algorithm for RL. In our evaluation, NeuralLinear performs well across datasets. Our insight is that, once the learned representation is of decent quality, being able to exactly compute the posterior in closed form with something as simple as a linear model already leads to better decisions than most of the other methods. We believe this simple argument is novel and encourages further development of this promising approach.
>
> 3) More generally, an interesting observation is that in many cases the stochasticity induced by stochastic gradient descent is enough to perform an implicit Thompson sampling. The greedy approach sometimes suffices (or conversely is equally bad as approximate inference). However, we also proposed the wheel problem, where the need for exploration is smoothly parameterized. In this case, we see that all greedy approaches fail.

---

> > ### Author Response · Authors · 2017-12-21
> > **Continued from above (due to max character count)**
> >
> > While collecting real-world datasets for a benchmark is challenging, the ones that we use are diverse. Some of them are not learnable or solvable (like Jester), while still of interest due to their practical applications (recommendation systems, in this case). For most datasets, we set the horizon to be the full size of the dataset, so it cannot be increased.  The regret appears linear because these are simply hard problems. Some dataset-dependent conclusions can be drawn: the Gaussian process does well on small datasets where it can handle a large proportion of the data, whereas constant-SGD performs much better on larger data.
> >
> > [1] Jasper Snoek, Oren Rippel, Kevin Swersky, Ryan Kiros, Nadathur Satish, Narayanan Sundaram, Mostofa Patwary, Mr Prabhat, and Ryan Adams. Scalable Bayesian optimization using deep neural networks. In International Conference on Machine Learning, 2015.

---

> > > ### Comment · AnonReviewer3 · 2018-01-13
> > > **The refinements made it a stronger submission.**
> > >
> > > The refinements made it a stronger submission. The authors also promised to release the code for reproducibility as Reviewer 1 recommended. I'm happy to change the score from 4 to 5.

---

### Public Comment · (anonymous) · 2017-12-20
**The description and connection to SG-MCMC literature might be improved**

Two comments:

1.  The description on the literature on Stochastic Gradient MCMC seems not accurate:

"A variety of methods have been developed to approximate HMC using mini-batch
stochastic gradients, These Stochastic Gradient Langevin Dynamics (SGLD) methods (Neal, 1994;
Welling & Teh, 2011) add Gaussian noise..."

SGLD is the mini-batch version of Langevin dynamics (1st order), while SGHMC is the mini-batch version of HMC (2nd order). Also, why (Neal, 1994) is cited for SGLD?

"Li et al. (2013) show that a preconditioner based on the RMSprop algorithm performs well
on deep neural networks."

It seems the correct reference is  "Li, C., Chen, C., Carlson, D.E. and Carin, L. Preconditioned Stochastic Gradient Langevin Dynamics for Deep Neural Networks,  AAAI 2016", NOT "Li, Ke, Swersly, Kevin, Ryan, and Zemel, Richard S. Efficient feature learning using perturb-and-map. NIPS Workshop on Perturbations, Optimization, and Statistics, 2013."

2. Connection between SGLD (in paragraph "Monte Carlo") and injecting noise on model parameters (in paragraph "Direct Noise Injection")

In terms of update rule, these two algorithms seem very similar: the update quantity in both consists of two parts: the gradient term and Gaussian noise term. Could the authors clarify the connections and compare them empirically if they are different?

---

> ### Public Comment · (anonymous) · 2017-12-20
> **Thanks for your feedback**
>
> We appreciate your interest and feedback on this paper.  Given the number of algorithms we compare, we unfortunately could not give a complete treatment of the background on each method.
>
> -"why (Neal, 1994) is cited for SGLD?"
> Neal is cited for SGLD because in his thesis he proposes Langevin dynamics (and HMC) for neural networks.  He experiments in Section 3.5.1 (page 103) with what he refers to as "partial gradients", which are gradient updates computed from single examples.  While he doesn't refer to it directly as stochastic gradient Langevin dynamics, we think it's fair to consider this the seed of the basic idea.  Teh and Welling also cite that work in the SGLD paper.
>
> - "SGLD is the mini-batch version of Langevin dynamics (1st order), while SGHMC is the mini-batch version of HMC (2nd order)."
> This seems like splitting hairs...  I appeal to Neal on the connection to Langevin dynamics.  From Neal, (Section 5.2 of https://arxiv.org/pdf/1206.1901.pdf): "The Langevin method: A special case of Hamiltonian Monte Carlo arises when the trajectory used to propose a new state consists of only a single leapfrog step."  Nevertheless, we will try to make our wording more precise.  A mention of SG-HMC seems warranted - however we didn't consider the methods because of the higher order terms involved.
>
> - "It seems the correct reference is  "Li, C., Chen, C., Carlson, D.E. and Carin, L. Preconditioned Stochastic Gradient Langevin Dynamics for Deep Neural Networks,  AAAI 2016", NOT "Li, Ke, Swersly, Kevin, Ryan, and Zemel, Richard S. Efficient feature learning using perturb-and-map. NIPS Workshop on Perturbations, Optimization, and Statistics, 2013."
> Thanks for finding that!  Yes, that should certainly be the other Li et al.
>
>
> - "2. Connection between SGLD (in paragraph "Monte Carlo") and injecting noise on model parameters (in paragraph "Direct Noise Injection"). In terms of update rule, these two algorithms seem very similar: the update quantity in both consists of two parts: the gradient term and Gaussian noise term"
> There are subtle (and tremendously interesting) connections between many of the methods presented here.  Parameter noise is also related to variational inference (if the posterior is assumed to be a diagonal unit variance Gaussian).  Yes, one can see how parameter noise (adding noise to the weights) can be thought of as related to SGLD (adding noise to the gradient updates).  It is e.g. easy to formalize the distinction between these two with a linear model  with squared error loss.  You will see that the scale of the noise added is very different and this is compounded with non-linearities.  However, the major difference is that SGLD is running a Markov chain while each sample from parameter noise is a draw from a diagonal Gaussian centered at the mean.  Thus SGLD can be shown to sample from the posterior (albeit under some strong assumptions) while parameter noise draws samples from a Gaussian approximation centered around the MAP estimate (the variance of which is left as a hyperparameter).
>
> "Could the authors clarify the connections and compare them empirically if they are different?"
> As stated above, they are rather different.  The paper is essentially an empirical comparison of the different methods and they behave tremendously differently empirically.

---

> > ### Public Comment · (anonymous) · 2017-12-20
> > **Thanks for your quick clarification**
> >
> > Personally, I agree with reviewer1, and believe this work could be very good contribution to the community for benchmarking the algorithms, if the implementation is of quality, reproducible and public available. Good luck.

---

### Author Response · Authors · 2018-01-10
**Main changes in the new version of the paper.**

A new version of the paper is now available. We updated the initial submission based on the reviews and the feedback provided in the additional comments.

The main changes to the initial version are the following:

- Implemented and tested an expectation-propagation algorithm (black-box alpha-divergence).
- Implemented and tested the sparse GP algorithm.
- Extended the algorithm description of variational inference methods, dropout, and sparse GPs. Also, we added the description of expectation-propagation methods.
- Extended the explanation of priors used by linear models.
- Extended the explanation of the Wheel bandit, and added explanatory plots to the main text.
- Extended the example that compares BBB with linear methods versus PrecisionDiag, and added two outcome plots to the main text.
- Updated and extended the experimental framework description, mainly metrics, regret, and hyper-parameter tuning.
- Updated and extended the discussion section, putting more focus on linking the statements to the empirical results in the tables.

- Added table that links names to specific algorithm configurations.
- Added table with the running time required by each algorithm and dataset.
- Added ranking column to cumulative regret table, where the mean ranking of each algorithm across datasets is shown. This way it is easier to parse the connection between the final big-picture conclusions and the empirical results.

- Removed cumulative and simple regret plots, given their low information content.

---

### Decision · Program_Chairs · 2018-01-29
**ICLR 2018 Conference Acceptance Decision**

**Decision:**

Accept (Poster)

**Comment:**

This paper is not aimed at introducing new methodologies (and does not claim to do so), but instead it aims at presenting a well-executed empirical study. The presentation and outcomes of this study are quite instructive, and with the ever-growing list of academic papers, this kind of studies are a useful regularizer.